# Dynamic modulation of inequality aversion in human interpersonal negotiations

Daniel A. J. Murphy[1,2], Jiaxin Xie[3], Catherine J. Harmer[3], Michael Browning[3,4] & Erdem Pulcu [3,4✉]

Negotiating with others about how finite resources should be distributed is an important aspect of human social life. However, little is known about mechanisms underlying human social-interactive decision-making in gradually evolving environments. Here, we report results from an iterative Ultimatum Game (UG), in which the proposer's facial emotions and offer amounts were sampled probabilistically based on the participant's decisions. Our model-free results confirm the prediction that both the proposer's facial emotions and the offer amount should influence acceptance rates. Model-based analyses extend these findings, indicating that participants' decisions in the UG are guided by aversion to inequality. We highlight that the proposer's facial affective reactions to participant decisions dynamically modulate how human decision-makers perceive self–other inequality, relaxing its otherwise negative influence on decision values. This cognitive model underlies how offers initially rejected can gradually become more acceptable under increasing affective load (predictive accuracy ~86%). Furthermore, modelling human choice behaviour isolated the role of the central arousal systems, assessed by measuring pupil size. We demonstrate that pupil-linked central arousal systems selectively encode a key component of subjective decision values: the magnitude of self–other inequality. Taken together, our results demonstrate that, under affective influence, aversion to inequality is a malleable cognitive process.

[1] University of Oxford Medical School, Oxford, UK. [2] St George's, University of London, London, UK. [3] University of Oxford, Department of Psychiatry, Oxford, UK. [4] Oxford Health NHS Foundation Trust, Oxford, UK. ✉email: erdem.pulcu@psych.ox.ac.uk

Negotiating with other people for one's own share of finite resources is an important part of human social-economic life. Recent historic events (e.g. "trade wars" between the United States and China, Brexit negotiations between the European Union and the United Kingdom) further highlight this importance, wherein the outcome of negotiations between only a handful of people will directly affect the lives of many others. However, there is limited work on computational and physiological mechanisms of such iterative social interactive decision processes relative to other domains, for example reinforcement learning.

The Ultimatum Game (UG) is a common behavioural economic measure which is used as an experimental probe of social interactive decision-making processes between two parties. In a recent study, we showed novel evidence to suggest that in iterative games *proposers* use a risky decision-making model to navigate around violating *responders'* fairness thresholds[1]. Proposers were shown to maximise their gains by choosing between Ultimatums based on their expected returns, while taking the probability of rejection into account. Rejecting an unequal proposal in the UG can be framed as a form of altruistic punishment[2–4]. Since both the proposer and the responder receive nothing when an offer is rejected in the UG, a responder's rejection sacrifices the amount of money offered at the expense of conveying an implicit message to the proposer that the amount offered had been unfair. Consequently, the responder's rejections in the UG violate the assumptions of the Rational Actor Model, which posits that any monetary gain is better than no gain[5] and all offers should be accepted.

Models of inequality aversion have been particularly influential in explaining "irrational" choices observed in UG field experiments[6,7]. A number of subsequent neuroimaging studies[8–11] showed neural correlates of human inequality aversion in regions associated with reward processing. On the other hand, studies using economic and evolutionary simulation models conceptualised responder behaviour in the UG in terms of reciprocity[12], dynamic learning[13,14], reputation building[15] and altruistic punishment[2]. Nevertheless, reporting summary statistics of average acceptance probabilities at different offer amounts is the most common approach. Consequently, how trial-by-trial computation of decision values underlying human responder behaviour take place in an iterative UG with an ecologically valid affective component has not been shown before.

During goal-driven social interactions, others' facial emotions provide us a window into how they perceive our requests, helping us to decipher their otherwise hidden valuation processes. Although humans detect subtle changes in others' facial emotions, the existing literature also suggests that people are prone to affective biases in facial emotion recognition[16–18], which may influence their decision strategies. Although a few decision-making studies have investigated the effect of proposers' facial emotions on human responder behaviour in the UG, how such affective information is integrated into subjective decision values in iterative games remains unknown. For example, a previous study showed that responders are consistently more likely to accept offers coming from attractive faces of the opposite sex irrespective of the offer amount[19], while a large-scale online study suggested that offers coming from proposers with smiling faces are more likely to be accepted relative to those coming from angry[20] or neutral faces[21]. However, a key limitation of these previous studies is the experimental approach: pairing affective faces randomly with different monetary offer amounts in repeated one-shot games. By this methodology, on each trial, participants are asked to respond to a stimulus which is intended to be completely independent of what they were presented with in the preceding trial(s). Considering that in daily social interactions people's facial emotions do not jump randomly from one affective

state to another, the previous experimental approaches would only have limited ecological validity in terms of capturing real-world human social interactive decision-making processes. In order to improve on this key limitation, we designed a novel sampling algorithm for an iterative UG task in which proposers' facial emotions and offer amounts were generated from two sliding windows with transition probabilities based on participant responses in the preceding trials (Fig. 1 and Supplementary Methods for transition probability tables). We had previously argued that this approach would break the trial-wise independence of experimental stimuli, allowing participants to experience a gradually changing social interactive decision-making environment based on their responses in the previous trials[22]. These modifications to the UG allowed us to probe participant choice behaviour using a large range of stimuli, involving both disadvantageous and advantageous/hyperfair offers[23]. We were also able to introduce a finer gradient on proposer's facial emotions relative to previous studies which made categorical distinctions, for example using happy versus angry faces. This protocol allows us to model two key influences on participant choice behaviour during social interactive decision-making: the magnitude of rewards and the proposer's affective state. Our a priori hypothesis regarding participant choice behaviour was that the proposers' facial emotions and the offer amounts should influence decisions to accept or reject offers. Although we anticipated that these two task components should interact, we did not have an a priori expectation about the direction of this interaction. In the following sections, using mathematical modelling of participant choice behaviour, we will describe computational mechanisms underlying how a proposer's facial expressions influence human responder behaviour by selectively modulating perceived self–other inequality in the UG.

In this study, we also collected a physiological measure, pupil size, which we think might give an insight about biological mechanisms that underlie social decision-making behaviour. Recent UG studies using simpler designs (e.g. focussing only on offer amount) and other physiological modalities such as EEG recordings demonstrated the utility of these measures in predicting participant choice behaviour[24]. Changes in pupil size in the absence of any experimental manipulation of external lighting conditions is known to reflect the activity of the central arousal systems[25]. Previous studies highlight a role for pupil-linked central arousal systems in human reinforcement learning (RL) and value-based decision-making, particularly when performed in dynamically changing environments[26–28]. Recent neurophysiology studies demonstrated that changes in pupil size reflect the firing rate of central norepinephrine neurons in the locus coeruleus[29,30] (LC). These studies provide a quantitative support for a number of converging theoretical[31] and experimental accounts of human behaviour[26–28,32], all implicating a role for the central norepinephrine (NE) system in guiding behavioural adaptations in dynamic environments, which can be assessed by measuring pupil size. Furthermore, previous work focusing on repeated one-shot games, suggested that unfairness arising from the discrepancy between self–other reward amounts in the UG engages autonomic arousal systems in humans (for example, cardiac response or skin conductance), also predicting participants' accept versus reject decisions[33,34]. Nevertheless, the role of pupil-linked central arousal systems during social interactive decision-making remains mostly unknown apart from one study conducted in a limited number of children ($N = 15$) which suggested that pupil dilation in response to viewing faces indexes familiarity with the face[35]. Considering that our UG task was specifically tailored to allow participants to experience an evolving interpersonal negotiation environment, we asked healthy volunteers ($N = 44$) to perform the novel experimental task while

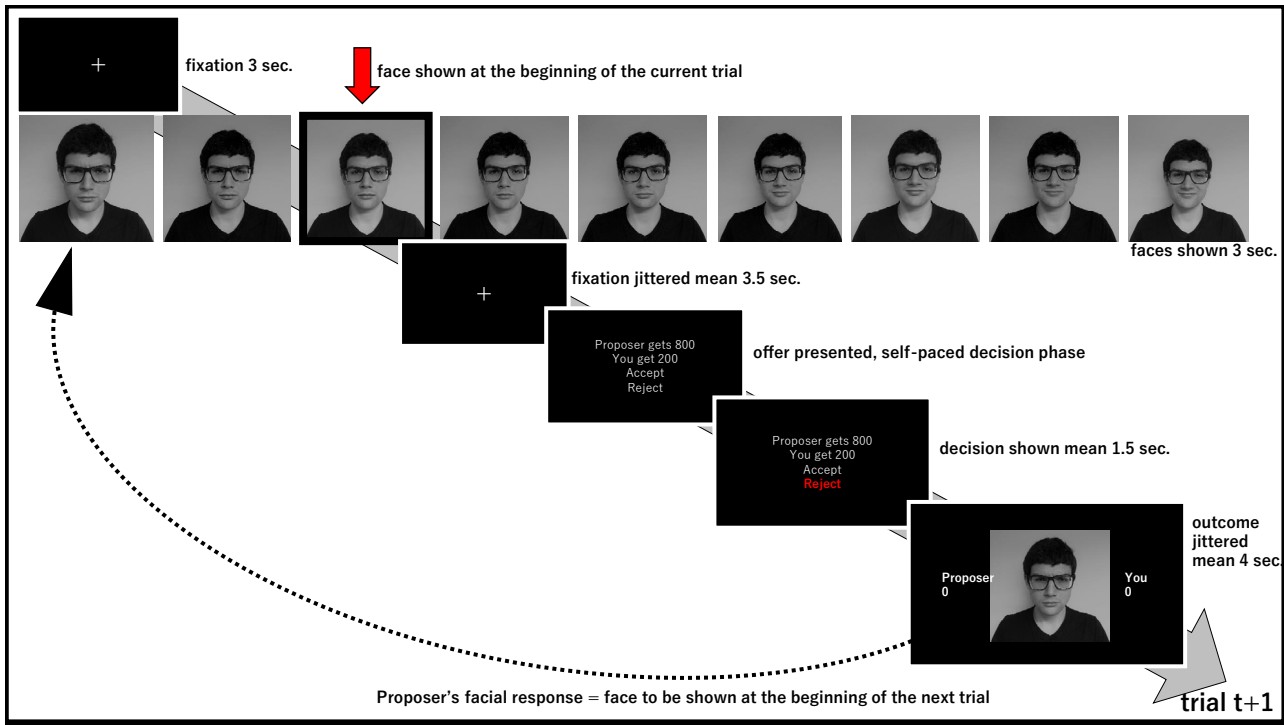

**Fig. 1 Experimental timeline of the novel Ultimatum Game task.** On each trial, the participant is presented with an affective face of the proposer/confederate (i.e. the red arrow) followed by an offer. In response to participant's acceptance or rejection, the proposer's facial emotion may stay the same or change to a neighbouring affective state based on predefined transition probabilities associated with each response type (e.g. the proposer is more likely to be happier if the offer is accepted). Similarly, based on other predefined transition probabilities, the offer on trial (t + 1) may stay the same or may be revised to a neighbouring offer amount based on participant response (e.g. the offer of 200 shown in the example above for trial (t) may increase to 250 on trial (t + 1), after the participant rejected the first offer). The predefined transition probabilities for facial emotions and offer amounts for the novel UG task are available in Supplementary Materials.

**Table 1 Demographic details of participants.**

| Measure | Mean (SD) |
|---|---|
| Age | 33.7 (11.30) |
| Gender | 68.18% Female |
| QIDS-16 | 8.11 (7.60) |
| Trait-STAI | 42.77 (13.52) |
| State-STAI | 33.43 (12.21) |

QIDS-16; Quick Inventory of Depressive Symptoms, 16 item self-report version. Trait/State-STAI; Speilberger State-Trait Anxiety Inventory. Note that scores of 6 or above on the QIDS-16 indicate the presence of mild depressive symptoms. The Trait/State-STAI has no standard cut-off scores.

undergoing pupillometry and explored the role of pupil-linked central arousal systems in social interactive decision-making. We expected that decision-making in an iterative UG should engage pupil-linked central arousal systems and encode components of the subjective decision value associated with unfairness which underlies participant choice behaviour.

## Results

**Participant demographics.** Forty-four participants recruited from the general public completed the UG experiment with pupillometry (further details of the experiment are available in the 'Methods' section). The demographic details of this cohort are summarised in Table 1.

**Human participants prefer fair monetary splits.** Participants initially rated 100 different Ultimatum offers presented in randomised order, on a 1-9 Likert scale indicating how much they

would like the proposed offer (all offers between £0.50 and £9.50 expressed in pence). These ratings allowed us to develop liking-based decision models for our main social interactive experiment. Participants' liking ratings monotonically increased from 50p (a highly unequal offer) to 500p (a 50/50 split), where they peaked. The liking ratings were lower for offers advantageous for the participant (>500p) relative to the 50/50 split, indicating an aversion to inequality (Supplementary Fig. 1).

**Nonlinear aspects of human facial emotion recognition.** After rating the Ultimatum offers, participants rated a range of their proposer's facial emotions also shown in Fig. 1, again on a 1–9 Likert scale from negative to positive. Overall, in this facial emotion rating task (FERT), participants' Likert ratings correlated highly significantly with the emotional valence of the proposers' facial expressions (average correlation coefficient $r = 0.869$, $p < 0.001$). However, participants were significantly better at detecting proposer's facial emotions when they were displaying positive compared to negative emotions (average $r$ values .821 versus .539, respectively; $t(86) = -4.55$, $p < 0.001$). This bimodality indicated the possibility of a nonlinearity in human facial emotion recognition that is more prominent for negative emotions (Fig. 2a). In order to capture this nonlinearity, which we thought should influence the way decision values are computed in the UG, we further analysed these ratings by exploiting the properties of a two-parameter exponential-logarithmic function (see 'Methods', Eq. (1)). Non-linearity in affective ratings was something that we anticipated from the planning stages of this experiment, and we intentionally used a 9-point Likert scale to assess this to enable use of this exponential-logarithmic function. However, it is important to highlight that we made no prediction

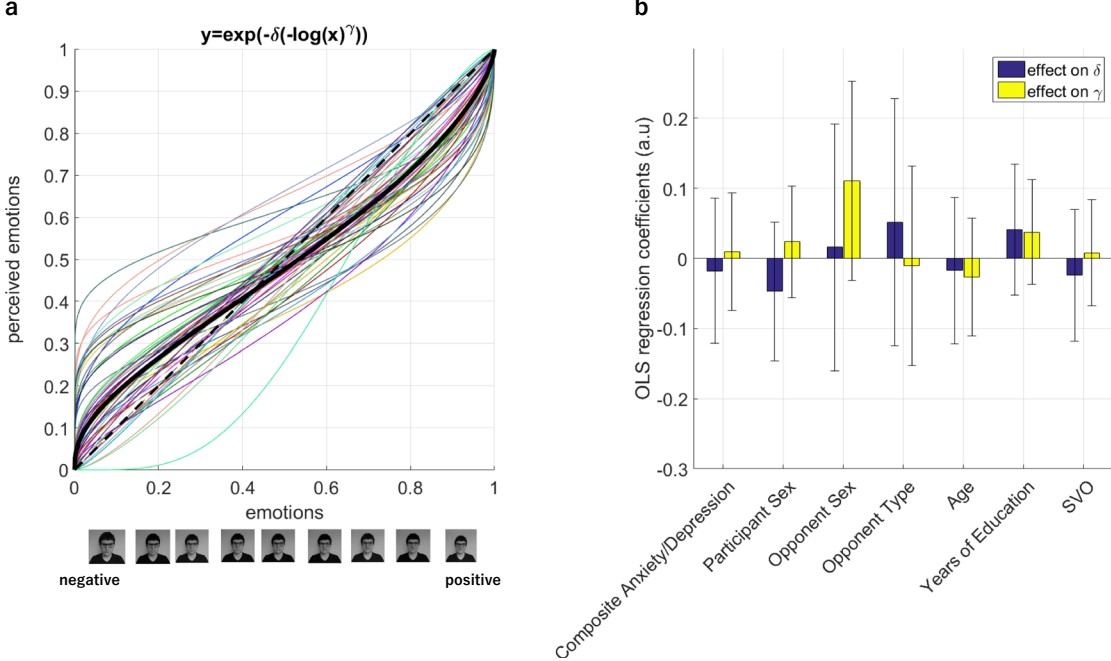

**Fig. 2 Results of the facial emotion rating task. a** Participants perceived their opponent's facial emotions nonlinearly. The x-axis shows the proposer's faces assembled from negative to neutral to positive (as in Fig. 1), and the y-axis represents participants perceived emotions according to the best-fitting nonlinear weighting model. The thick black line designates the population mean. **b** The model parameters [δ, γ, see Eq. (1)] were not influenced by factors such as symptoms of depression and anxiety, or participant and proposer sex. Composite depression and anxiety scores were established for each participant by linearly transforming (i.e. summing up scores from each domain within subjects) the z-scores of depression, and state and trait anxiety measures to avoid collinearity in the model, as these measures were highly correlated with each other in this cohort of non-clinical volunteers (all $r(42) > 0.69$, all $p < 0.001$). Opponent type designates whether participants were told they would be playing against a human confederate or a computerised proposer. SVO: Social Value Orientation, which is a continuous measure defining one's degree of prosociality. All regressors were normalised before model fitting to allow a direct comparison. Error bars denote 95% CI. Variance inflation factor ($1/(1-R^2)$) computed for this model ranged between 1 and 1.06, where values greater than 5 would raise concerns for multicollinearity between regressors. The same set of regressors were used for the analysis reported in Fig. 5.

about the exact trajectory of this nonlinearity due to lack of previous work. This novel analysis approach confirmed that participants perceived their proposer's affective states nonlinearly, albeit with some individual variability (Fig. 2a). We further explored whether the parameter estimates from this model could be influenced by participants' own mood states (i.e. factors such as symptoms of depression and anxiety, as measured by validated clinical questionnaires). This analysis did not suggest the continuous or categorical variables that we considered in the regression model significantly influenced estimates of the model parameters (Fig. 2b). Factors such as opponent type (i.e. whether the proposer was a confederate or a computerised opponent) or opponent sex did not influence the parameter estimates either, indicating that the nonlinearity captured in participants ratings was not a direct result of the way proposers might have expressed their emotions while their pictures were taken initially.

**Proposer's facial emotions influence choice behaviour in the UG.** We first concatenated all behavioural data from each participant and binned average acceptance probabilities for all combinations of all proposers' facial emotions and offer amounts (Fig. 3a) and analysed this data with an ordinary least squares (OLS) regression model. This analysis suggested a main effect of facial emotion (based on $t$-tests on OLS coefficient estimates, $t(43) = 8.43$) and a main effect of offer amount ($t(43) = 8.85$). The effect sizes for these main effects were $d = 1.262$ and $1.325$, respectively. These findings also confirm that the task successfully probed both of these components. We also identified a significant facial emotion × offer amount interaction ($t(43) = 3.70$) influencing

participants' probability of accepting an offer (all $p < 0.001$, Fig. 3b). This indicates that people were more likely to accept unfair offers if the proposer's facial emotion was more positive, agreeing with previous results from one-shot games[20]. However, it is important to point out that these simple behavioural effects give a bird's eye view but cannot account for the iterative/dynamic nature of our social interactive decision-making task. For completeness, we report the average number of trials participants spent in each state-space during the UG experiment (facial emotion × offer amount) in Supplementary Fig. 2.

**In iterative games history of recent events continues to influence participant choice behaviour.** One of the main features of our experimental design was that participants responded to offers generated based on their responses in preceding trials. We used this approach in an attempt to increase ecological validity compared to the trial-wise independence of stimuli commonly employed in previous studies. To demonstrate this effect, we investigated how stimuli shown in preceding trials (i.e. $n$-1th to $n$-3th), as well as participants' previous decisions, influenced their choice behaviour on the current trial (the $n$th trial) using a logistic regression model. We think this analysis approach, focussing on the effects of only the previous trials on the current choice, would complement the model-free behavioural results focussing on acceptance probability reported in the preceding section. There were four regressors in this model: proposer's facial emotion, the offer amount, a facial emotion × offer amount interaction term, and the participant's choice. This analysis suggested all regressors from the $n$-1th trial significantly influenced

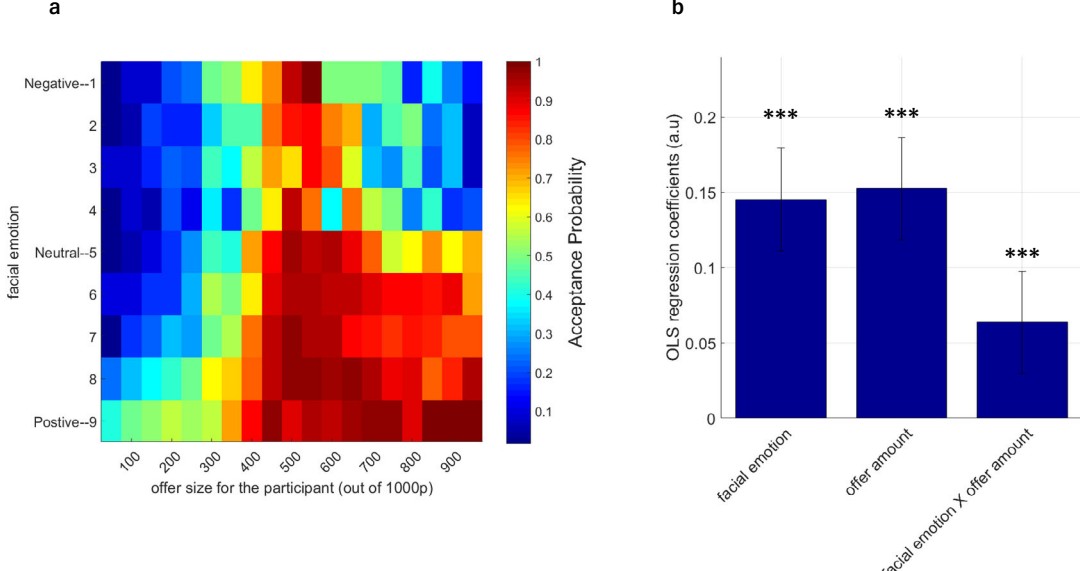

**Fig. 3 Behavioural results from the Ultimatum Game. a** Participants' average acceptance probabilities across all possible combinations of proposer's facial emotion (y-axis) and offer amounts (x-axis) represented as a heat map. The data shown in the heat map is concatenated across all participants and all conditions. Colour bar shows the probability of accepting an offer. Changes in the colour gradient in the heat map suggest that unfair offers coming from positive faces were more likely to be accepted relative to offers associated with negative facial emotions, even if the negative-emotion offer amount is advantageous to the participant. **b** A formal OLS regression analysis conducted on the acceptance probabilities indicated a significant main effect of facial emotion, a main effect of offer amount and a significant facial emotion by offer amount interaction term influencing participants' probability of accepting an offer (***$p < 0.001$). Error bars denote ±1 SEM.

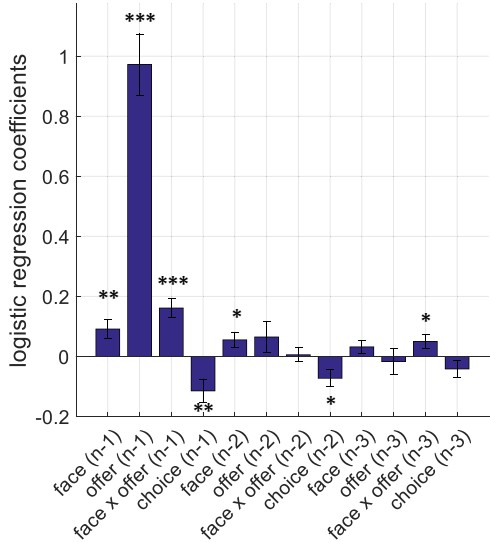

**Fig. 4 Logistic regression analysis of participant choice behaviour.**
Coefficient estimates from the logistic regression model fitted to participant choices on the current (*n*th) trial indicate that all regressors from the *n*-1th trial significantly influence participant choice behaviour (***$p < 0.001$, **$p < 0.01$). The coefficient estimates for the influence of previous choices were all negative, indicating that offers on the current trial are more likely to be accepted if offers on previous trials were rejected. The influence of these variables decayed down the trials. Error bars denote ±1 SEM. The average variance inflation factor computed for these 12 regressors was 1.0997.

participant choice behaviour on the *n*th trial (all $|t(43)| > 2.84$, $p < 0.01$), with the offer amount on the previous trial being the most significant influence ($t(43) = 9.56$, $p < 0.001$; Fig. 4). The influence of these variables on participant choice behaviour decayed with preceding trials. These results are also in line with

the general notion that human choice behaviour in social interactive decision-making games can be defined in terms of *n*-1 (i.e. memory-1) conditional-probabilistic strategies[36].

**Proposer's emotions dynamically modulate perceived self–other inequality**. We considered a number of computational models to describe participant choice behaviour in the iterative UG task (see 'Methods' for model descriptions). Computational modelling allows us to decompose the underlying mechanisms of the main effects that we demonstrated in the preceding sections through a dynamic analysis approach better suited to the iterative nature of the experimental task.

The best-fitting model (Model 10), which was able to correctly predict participant choices in 85.8% of the trials (i.e. average percent of trials in which individual participant's and model's choices are in agreement), assumed that nonlinearity in the way the proposer's facial emotions are perceived (as shown in Fig. 2a). In the best-fitting model, perceived emotions modulate participants' perception of inequality[7] in the UG, and relaxes/neutralises the negative valuation associated with it (Supplementary Materials, Eqs. 1 and 9). Our proposed mechanism for this influence was represented by a parabolic relationship, meaning that increasingly negative or positive emotions displayed by the proposer would have a similar effect on perceived inequality during social interactive decision-making. In descriptive terms, this would mean that if the proposer insists on an offer amount which is rejected iteratively by the participant and responds to the participant's rejections by displaying increasingly negative emotions, at some point the proposer might overcome the participant's negative valuation associated with perceived self–other inequality. This means that under affective load, previously rejected offers gradually become more acceptable to the responder (i.e. acceptance probability would increase). However, it is also important to highlight that the effect of facial emotions may not be evident in all participants as it would also depend on the strength of the inequality aversion (both of these processes have their own nonlinear trajectories as shown in Fig. 2b (facial emotion

recognition) and Supplementary Fig. 1 (liking ratings may be interpreted as a proxy for inequality aversion)), such that even if the acceptance probability increased under affective load this would not necessarily mean a behavioural switch that would be applicable for all conditions. The effect of previous rejections on decision to accept the offer in the current trial can also be observed in negative logistic regression coefficients reported in Fig. 4, complementing these results.

The best-fitting model can also flexibly adapt to other situations that might occur during the game. For example, in situations where the proposer's facial expressions are neutral, the influence of perceived affective state is diminished (Eq. (9)), while perceived inequality becomes relatively more important in the decision-making process. In situations in which the proposer displays increasingly positive emotions, although this condition may be hypothetical and would not occur too frequently in the current design, the model can reduce the overall influence of perceived inequality on subjective decision-values, accounting for compromise behaviours (e.g. "the offer is unfair, but if I accept it, it would at least make the other side happy"). Bayesian model selection metrics[37] supported these assertions, showing that the model in which facial emotions selectively act on the inequality term (Model 10)—but not on either the self-reward amounts term (Model 8) or modulating both the self-reward and the inequality term independently (Model 9, Supplementary Fig. 3)—fits better to explain participant choice behaviour. The parabolic (Model 10), rather than exponential (e.g. Model 6, Supplementary Fig. 3), shape of this affective influence indicates that a proposer's increasing negative or positive facial emotions would act on the inequality term in a similar manner, while neutral facial emotions would have negligible influence on participant's perception of inequality. These results were validated in an independent cohort (n = 25) which confirmed that reported effects or the best-fitting model were not influenced by task instructions or completing rating tasks prior to the main experiment (see the section on Control experiment). Further analysis of trials mispredicted by the best-fitting model did not show any significant main effect of the proposer's facial emotions or offer amounts (Supplementary Fig. 4), indicating that the model does not fail systematically in converting these input stimuli into subjective decision-values. However, there was a significant facial emotion × offer amount interaction in the mispredicted trials (t(43) = −3.2452, p < 0.01). Negative regression coefficients on this interaction term indicate that the model struggled to account for participant choice behaviour at the extremes (e.g. unfair offers coupled with very positive faces). Considering the close agreement between the raw (Fig. 3) and the simulated data based on the best-fitting model (Supplementary Fig. 6b, c) we did not further explore the decision model space by assigning separate parameters for advantageous versus disadvantageous types of inequality. This was an intentional choice to limit the model space to avoid overfitting to the current dataset, particularly considering the best-fitting model had good predictive accuracy, stability and was able to recapitulate human behaviour (Supplementary Fig. 5).

One possibility that we did not consider so far is that our participants could engage with the task with a "risk of missing out" in mind, such that repeated rejections could put pressure on participant choice behaviour and accept subsequent offers, so that they would be paid at least some additional payment on randomly selected trials (note that there would not be any additional payment if participant rejected the offers on all the randomly selected trials). According to this framework, the participants' expected payment at the end of the game would be a function of their average acceptance rate multiplied by the average offer they experienced during the game, as they could not know which trials would be selected at the end of the experiment. If participants engage with the task and were influenced by the

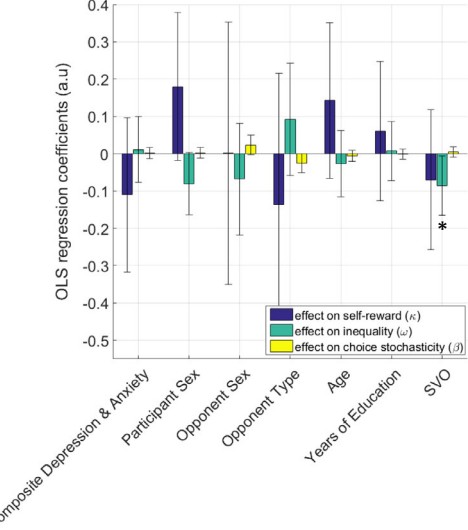

**Fig. 5 Prosocial individuals perceive self–other inequality more negatively in the UG.** Coefficient estimates from an OLS regression model fitted to parameter estimates of the best-fitting computational choice model (i.e. Model 10) suggested that people with higher social value orientation will perceive self–other inequality more negatively (*p < 0.05). A generate-recover simulation analysis (50 iterations) based on stochastic choices generated by this model demonstrates the stability of the estimated parameters (Supplementary Fig. 5). Error bars denote 95% CI. Note that these regression coefficients do not need to be subjected to further correction for multiple comparisons. All regressors were normalised before model fitting to allow a direct comparison.

proposer's facial emotions according to the parabolic model that we proposed, there should be a negative correlation between the fitting of this model and participant's expected outcome described above, as they would be swayed away from this expected outcome calculation by the influence of proposer's facial emotions according to the best-fitting model. We conducted an exploratory analysis on this relationship and observed that to be negative, which might serve as complementary evidence in favour of our proposed model (main experiment, r(43) = −0.21, p = 0.17; combined cohorts including the control experiment r(68) = −0.29, p = 0.016; Bonferroni corrected over 2 comparisons).

Similar to the OLS analysis that we reported for facial emotion recognition parameters (Fig. 2b), we analysed which variables influence the parameter estimates of the best-fitting choice model in the UG. This analysis suggested a significant relationship between participants' social value orientation (SVO, as measured by the SVO Slider Measure[23]) and parameter values of the inequality term estimated individually from each participant's choice behaviour, meaning that people with higher SVO scores (i.e. people with more altruistic tendencies) perceive self–other inequality more negatively (t(43) = −2.166, p = 0.036, Fig. 5). Here, it is critically important to highlight that the opponent-type regressor which determines whether the participants played against a human, or a computerised opponent did not have any significant main effect on human behaviour or parameter estimates. It is possible that recent advances in AI mimicking, and even excelling, human behaviour in competitive games[38–40] might have an indirect effect on these results (i.e. human participants may find it easier to attribute human features to the computerised opponents relative to studies conducted in earlier decades). In the context of social reinforcement learning, a recent study using abstract fractals to represent human versus non-human targets in a two-option forced-choice paradigm suggested that participants relied on trait generosity when learning about

humans, whereas relying on reward rate for non-humans (i.e. slot machines)[41]. Intriguingly, in our previous work focusing on the proposer behaviour in the UG[1], we have demonstrated that participants rely on SVO-related learning models (i.e. similar to trait generosity related models mentioned earlier) for understanding the decision tendencies of computerised responders that are represented by chess icons. It is also possible that iterative social interactive games may be reducing participants' understanding of such differences. Although a strong a priori assumption would dictate that social interactive decision-making should be fundamentally different than non-social forms of value-based decision-making (including interactions with computerised algorithms), as mentioned, these results are globally in line with not only our previous work[1], but also work from other influential labs studying these decision problems with or without real human agents[42]. Although effect sizes for the main task components were robust (see in the preceding sections), it is also possible that our sample size was not large enough to detect some of these relationships (also applies to results reported in Fig. 2b).

**Pupillometry results**. We analysed participants' pupil response during the decision stage of the UG task (i.e. the window from the presentation of the offer until the end of outcome presentation) with an OLS regression model to explore the extent to which regressors generated under the best-fitting computational model correlated with pupil size during the UG experiment (see 'Methods' for the details of the pupillary regression model, Eq. (11)).

One possible caveat that we evaluated quantitatively before fitting the pupillary regression model is the correlation between proposers' perceived facial emotions and the offer amounts given to the participant. Normally, in general linear models, one would try to decorrelate these regressors as much as possible to evaluate their unique pupillary or neural signatures. However, in the case of our task, the computerised strategy was designed to display positive faces with higher probability if the offers were accepted (as should logically happen in interpersonal exchanges in real life). In the extreme case of a proposer's facial emotions and offers being completely decorrelated, the computerised proposer may no longer be perceived as a human proposer (e.g. proposer reacting to participant's decisions by showing unexpected and/or unrelated/random emotions or offers). The histograms of correlation coefficients between proposers' perceived facial emotions and decision-values generated under the best-fitting model for each participant are shown in Supplementary Fig. 6a (correlation coefficient $r$ (mean ± SD) = 0.058 ± 0.26). This demonstrates that our experimental manipulation was able to deal with collinearity issues adequately, something which would otherwise compromise the quality of the pupillary multiple linear regression analysis.

In a similar manner to our previous study[1], we also asked our participants a number of questions related to how they felt about their proposers to make sure that the computerised strategy was perceived as "human-enough" (full set of questions given in Supplementary Fig. 6b legends). On average, participants were able to identify more than one person from their social circles who would make offers and display affective reactions in a similar manner to the computerised proposer (response to Q3; mean (±SD) = 3.82(±0.40); t(43) = 7.12, $p < 0.001$), reassuring that our experimental manipulation was successful in terms of the computerised proposer adequately mimicking human behaviour while keeping the correlations between perceived faces and offer amounts within an acceptable range.

To be able to extract a cleaner signal from our subjective decision value regressors, we wanted to regress out as much of the unaccounted variance from pupil size timecourse as possible. We thought that regressors which quantitatively defined how the decision-making environment changed (i.e. environmental volatility and environmental noise[43]), as well as participants' response to those changes (i.e. surprise) would adequately reduce variance unaccounted by our key regressors. These components were estimated from the raw stimuli (i.e. the offer amount and facial emotion valence on each trial) by a recursive Bayesian filter[43] that can estimate the generative statistics of the stimuli uniquely for each participant, and helps with objectively quantifying how the social interaction environments change (also see Supplementary Methods for further details). Such changes also depend on participant choice behaviour, meaning that each participant experienced a unique sequence of stimuli. Regressors defining environmental volatility, noise and surprise response did not lead to any statistically significant pupil dilation (Supplementary Fig. 7), suggesting that higher-order statistics of the social interactive decision-making environment did not engage the pupil-linked central arousal systems.

**Pupil size encodes the magnitude of self–other inequality prior to decision onset**. Under the best-fitting model, subjective decision values were a function of 3 regressors with trial-wise variability, namely the self-rewards (Fig. 6a), perceived facial emotions of the opponent which modulate how negative self–other inequality is perceived (Fig. 6b) and the magnitude of the self–other inequality (Fig. 6c). We were mainly interested in the pupillary correlates of these regressors.

Subsequent analysis performed on the average pupillary regression coefficients binned at each second after offer presentation, by one-sample t-tests from baseline, indicated that pupils dilate more in reaction to the magnitude of inequality which peaks prior to decision onset (peak response between 0 and 1000 ms, t(42) = 2.893, $p = 0.006$, Fig. 6c) and decays continuously after the decision onset.

## Discussion

In the present work, we describe value computations underlying human responder behaviour in an iterative UG task which also involved participants observing the proposers' affective reactions. Our results suggest that human participants exhibit perceptual biases in facial emotion recognition which are represented non-linearly (Fig. 2a). We show that proposers' facial emotions significantly influence participant choice behaviour in social interactive decision-making (Fig. 3a, b). The influence of the proposer's facial emotions on participant choice behaviour is confirmed by different methods of analysing the behavioural data (i.e. both logistic regression and computational model-based). Computational modelling results demonstrate that human responders use an inequality-aversion-based model which is dynamically modulated by the proposer's facial emotions. We show that parameter estimates for the inequality term from the best-fitting computational decision-making model were significantly negatively correlated with participants' social value orientation (SVO). This captures the intuition that people with higher prosocial tendencies perceive self–other inequality more negatively and are less likely to accept unfair offers (Fig. 5). Computational modelling of participant choices further revealed that opponent's affective reactions dynamically modulate participants' perception of inequality, relaxing its negative influence in a *parabolic* shape (i.e. Model 10). Participants became more likely to accept unfair offers, irrespective of whether they were advantageous or disadvantageous, when their rejections were confronted with iterative positive or negative affective response from

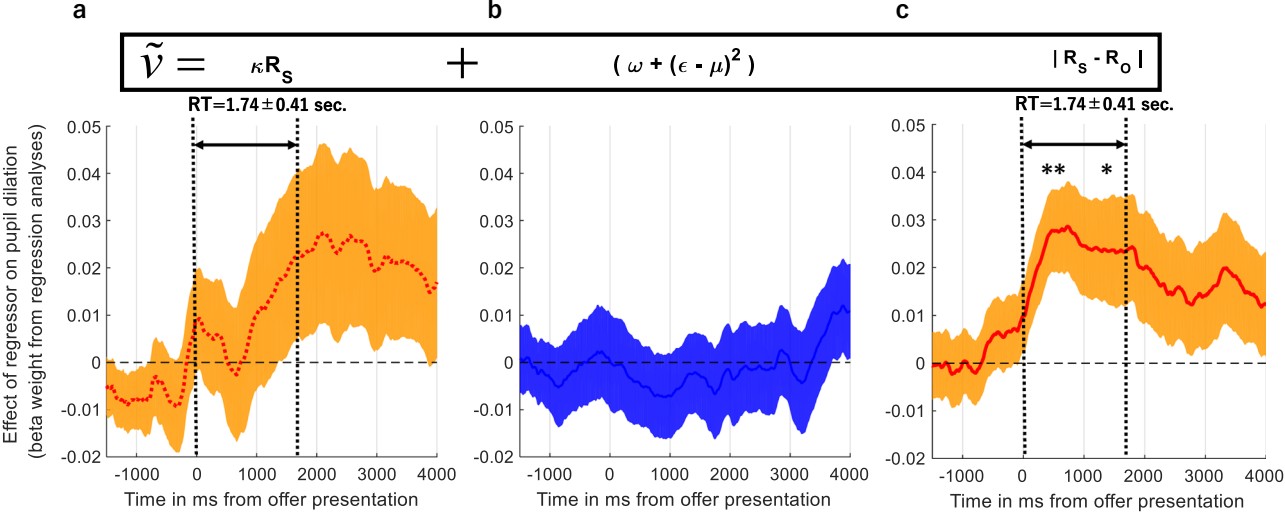

**Fig. 6 Pupillary signal of subjective decision-values computed under the best-fitting model. a** Coefficient estimates from the pupillary OLS regression model correlating with self-reward values (i.e. the orange line and shading) (**b**) perceived emotions of the opponent which dynamically modulate the inequality term (**c**) the magnitude of self–other inequality. These panels illustrate the components of the subjective decision values. Computed under the best-fitting model as shown above the figure in a black frame (also in Eq. (9)). Pupil response to inequality peaks prior to decision onset (the window marked with dashed lines designating the decision reaction time (RT)) and indicates that pupil dilates more to offers associated with greater inequality ($**p < 0.01$, $*p < 0.05$). Error shading denotes ±1 SEM across all panels. All regressors were normalised ahead of model fitting to allow direct comparisons.

the proposer (also partially illustrated in the model-free faces × offer amount interaction effect, Fig. 3).

This model can flexibly account for human behaviour in a number of social interactive decision-making scenarios, for example compromise behaviours to emerge in interpersonal negotiations under affective load, demonstrating that human inequality aversion is a malleable construct. Exploring the correlates of the regressors generated by the best-fitting computational model suggested that the inequality component of subjective decision values that guide participant choice behaviour engage the pupil-linked central arousal systems prior to decision onset. To the best of our knowledge, these results constitute the first line of evidence giving a detailed account of (i) how opponent's affective responses influence value computations in social interactive decision-making; and (ii) the role of the pupil-linked central arousal systems in these value computations.

Among behavioural economic games, the UG is one of the most commonly used zero-sum game. This is because for any accepted offer, gains are mutually exclusive between the proposer and the responder (i.e. if the proposer gets more of the amount to be split, the responder gets less). The linear payoff structure and sequential decision-making nature of the UG makes it a good candidate for computational modelling compared to other nonzero-sum games like the Prisoner's Dilemma (PD). The UG is known to have a strong non-monotonicity, which indicates that people's acceptance probability does not increase monotonically with increasing self-reward magnitudes[44]. This forms the basis of nonlinear aversion to inequality commonly observed in human decision-makers (Eq. (6), $\omega$ parameter). We demonstrated that our participants also behaved in this manner, as indicated by results from the analysis of choice behaviour (Fig. 3a) and liking ratings (Supplementary Fig. 1). We further demonstrate both in our model-free (Fig. 3) and model-based analyses that the affective content of the proposers' facial expressions modulates these decision processes. One of the most important contributions of our current work is the suggestion that the nature of this affective influence is better accounted by a parabolic model relative to other models that we considered.

The present results complement the findings of a recent work in which we described a computational model accounting for

human proposer behaviour in the UG[1]. In these two studies, focussing on value computations underlying proposer[1] and responder behaviour separately have allowed us to restrict the vast model space and identify suitable models accounting for human social interactive decision-making. Future studies should go one step further than these preliminary reference points that we identified in the UG decision model space, and independently validate and replicate risky decision-making models accounting for proposer behaviour versus inequality aversion models accounting for responder behaviour in a two-person interactive experimental design. Although, in our own evaluation, the current experimental approach tackles many of the limitations of previous UG studies, it is still far away from a naturalistic two-person exchange. One limitation of our study which may be important to highlight is that our experimental design did not allow the offers to stay stable for reasonably long sequences to be able to dissociate the effect of the change to facial expressions on the inequality term further (i.e. average maximum sequence length in which the offers stayed the same per participant was 3.5 trials ([min, max]: [2, 11]). As a result, our model proposes that human inequality aversion is malleable through a parabolic influence, but this proposal emerges as a winning model from the whole stimuli space rather than specifically confirming this hypothesis in a task design in which the offers remained constant and only the facial emotions are manipulated. Consequently, we think future studies should test the suitability of our proposed model in iterative UG tasks perhaps with different rules (e.g. subsequent offers generated based on previous decisions but coupled with randomly generated facial emotions) or ideally involving a genuine two-person interaction that would also address one of the key limitations of the current work, namely disguising an ad hoc probabilistic sampling algorithm within an experimental design which involves human confederates.

The extent of the decision model space has been one of the key factors limiting the number studies reporting computational models of choice behaviour in social interactive games[45–47], as the difficulty associated with modelling recursive theory of mind (ToM) processes is making social interactive decision-making a topic "too hot to handle". A recent neurophysiology study in

monkeys has shown that it is possible to bypass some of these difficulties by modelling the proposer's future decisions in a binary format (i.e. cooperate versus defect in PD) to illustrate underlying neural correlates of ToM processes[48]. However, these novel approaches require neural data with very high temporal precision, something that has not yet been implemented in human participants. As a result, and despite the fact that humans are social animals, our understanding of social interactive decision-making processes in healthy and patient groups are lagging far behind other well-defined cognitive processes such as reinforcement learning (RL)[27,28,49,50]. In fact, RL models have been exploited to describe choice behaviour in one-shot UG where participants played against different opponents in individual trials, however, this approach necessitated the participants learn the norms of a population of proposers to allow models to make use of prediction errors arising from violations of the population norms[51]. Nevertheless, we demonstrate that an iterative choice model converting reward and affective information into decision values can account for wide majority of iterative human social interactive decision-making (model predictive accuracy ~86%) even without recursive ToM modelling. One might argue that our modelling approach for the current study ignored trial-by-trial learning processes which might be taking place during social decision-making. It is possible that incorporating these additional cognitive processes could further improve the predictive accuracy of our proposed model. In our previous work focusing only on offers but not offer/facial emotion combinations in the UG, we had used various Bayesian observer, SVO and RL models to account for learning processes with which proposers' understand the responders' binary accept versus reject decisions across 180 trials. However, learning how proposers would change their behaviour across an offer by facial emotion grid space in the face of rejection from the participant is much more complex than learning binary accept versus reject decisions. Considering that the current experiment also involves proposers' affective reactions (in total 9 faces), one would ideally need around 1500 trials to be able to adequately model learning processes. In our case, this approach was not feasible to implement. This limitation is also illustrated by the distribution of trials as shown in Supplementary Fig. 2, each state-space does not have enough number of trials to model distinct learning curves. These limitations have also been communicated by other key computational modelling work concerning social interactive decision-making[46,47].

One of the key rating tasks which informed our modelling of participant choice behaviour in the UG evaluated participants' perceptions of proposers' facial emotions. We modelled participants' ratings on a Likert scale by exploiting the properties of a two-parameter weighting function. Although our modelling approach revealed individual variability in these perceptual biases, the parameters of this model did not correlate significantly with symptoms of depression (QIDS range: 0–27) and anxiety in this non-clinical cohort (Fig. 2). These results are not in line with predictions based on some of the previous studies in depressed groups which used the FERT[16,17]. Consequently, it may be worthwhile to highlight some of the experimental differences between the traditional FERT and the facial emotion rating task that we used in the current study. In the traditional FERT, participants are presented with affective faces for a very brief period, approximately for 800 ms; and they are asked to label faces into one of seven different emotional categories: sad, angry, happy, surprise, fear, disgust and neutral. Brief stimulus presentation duration does not allow too much time for participants to decode the individual parts of the face while forming a judgement about its affective content. The faces are presented in an oval frame cutting out some of the facial features such as hair and ears

(approximate size 56.25 cm$^2$). It is possible that these aspects of the FERT create ambiguity and an information gap which probes affective biases more strongly in clinical groups relative to non-clinical volunteers. On the other hand, our rating task was self-paced (i.e. ratings were made while viewing the faces, RT (mean ± SD) = 2.90 ± 1.04 s). It is possible that accurately rating affective faces on a scale may be more challenging than categorising them into predetermined labels (e.g. happy, sad, fear etc.). Speculatively, ability to categorise other's emotions quickly and accurately may have survival value similar to rapid fight-or-flight decisions. It is possible that evolution favoured agents who are better at recognising other's facial emotions, as in the case of negative emotions, different affective states (e.g. sad vs angry) often cue distinct action tendencies. In the rating task, the faces were presented in a rectangular frame similar to a passport picture occupying a 102.7 cm$^2$ area. These modifications allow participants to gather more information about the affective content of the faces, reducing ambiguity and increasing ecological validity of the stimuli and the ratings (e.g. the fact that in real-life social interactions people have reasonably long time to observe others' facial emotions, see a recent perspective for a critical appraisal of existing task designs in cognitive neuroscience[52]). Therefore, in future studies, it would be very important to gather large-scale data to validate the affective content of the facial expressions used in the current study, which can help understanding how symptoms of depression and anxiety influence facial emotion recognition in the wider population. While our current modelling results suggest that facial emotion processing (i.e. evaluating intensity and valence) and social interactive decision-making processes are not systematically impaired by increasing symptoms of depression and anxiety in a group of non-clinical volunteers, it would be informative to implement the current experimental design in clinical groups, such as patients with major depression, in which facial emotion recognition and social decision-making processes were shown to be affected in previous studies[9,22,53,54].

A number of previous studies reporting computational mechanisms which under learning and decision-making in dynamically changing environments have interpreted phasic changes in pupil size as an index for the firing of the central NE neurons[26–29]. In the current work, we showed that pupil dilation correlates significantly with a key component of subjective decision values that selective relates to the magnitude of self–other inequality, prior to decision onset. Although the relationship between the firing of NE neurons and pupil dilation is often mentioned in the literature, it is important to acknowledge that pupil dilation in response to various task components may also be under the influence of cholinergic and serotonergic (5-HT) activity in the brain (see a more detailed discussion in Faber[55] and Muller et al.[32]). For example, there is a rich body of literature showing the effects of 5-HT in social decision-making[3,11,56] and facial emotion recognition[18,57–59], and previous work has shown that pharmacological agents acting on the 5-HT system also modulate pupil dilation[60,61]. One of the more recent social decision-making studies investigating the effects of agents acting on the 5-HT system across two repeated one-shot studies (both $N = 20$), used a trial-by-trial analysis approach implemented in generalised estimating equations to show that both MDMA and psilocybin reduced rejection of unfair offers[62]. Teasing out the influence of these neurotransmitters on pupil dilation during social interactive decision-making ideally requires experimental pharmacology methods and should be tested in future studies to further understand mechanisms of causality. The current experimental approach could also be useful in teasing apart neural responses associated with information processing during social interactive decision-making, considering existing literature clearly demonstrates that facial emotion recognition and social

decision-making activate fronto-striatal and limbic regions[4,8,47,56,63–65]. The main strength of the current experimental design is that it probes facial emotion processing and social decision-making domains simultaneously with a relatively fine gradient (by generating stimuli probabilistically from two sliding windows, 9 different facial emotions × 19 different offer amounts, in total 171 unique stimulus combinations, Fig. 2a), therefore it can be used as an experimental probe for addressing these mechanistic questions about information processing during social decision-making. We predict that the correlates of the subjective value regressors from the stochastic choice model (Fig. 6) should be encoded by the fronto-striatal and limbic regions.

Taken together, our key results demonstrate that, under affective load, aversion to inequality in human participants is a malleable cognitive process. Compared with previous work that showed 50% of responders systematically adapt their choice behaviour when statistics of expected offers are varied in different UG task blocks, our model proposes a mechanism that accounts for dynamic changes in aversion to inequality[66]. We show that central arousal systems—reflected in pupil size—are involved in value computations during social interactive decision-making, and track the opponent's affective responses. We show that these perceived affective responses dynamically modulate the perceived inequality of an offer, and therefore its probability of acceptance. These findings may have important implications for understanding the cognitive processes that underlie suboptimal outcomes (e.g. compromise behaviours and settling down for an unfair split in interpersonal negotiations) in social interactive decision-making.

## Methods

**Participants**. Forty-four participants were recruited from the local community via advertisements. Potential participants who had a history of neurological disorders or who were currently on a psychotropic medication were excluded from the study. In order to increase the overall generalisability of study results, eleven confederates were recruited from the staff of University of Oxford Department of Psychiatry. The methods were performed in accordance with relevant guidelines and regulations and approved by Oxford Central University Research Ethics Committee (CUREC). All participants and confederates provided a written informed consent prior to taking part in the study. We have confirmed consent to publish the images of the confederate, as shown in Fig. 1.

The confederates attended to two testing sessions. In their first visit, they were instructed to display different emotions (i.e. 9 different from negative to neutral to positive) while the pictures of their faces were taken. When their pictures were taken, the confederates were instructed to start with a neutral facial expression and then display increasing positive emotion in four increments and finally display increasingly negative emotion, also in four increments. These pictures were later used in the facial emotion rating task and the Ultimatum Game (UG, Fig. 1). In order to increase the ecological validity of these facial emotion stimuli, we did not instruct confederates to display specific emotions (e.g. disgust), but allowed confederates freedom to express negative/positive emotions as they would in their personal lives.

**Procedures**. Before the experimental tasks, participants completed the Spielberger State-Trait Anxiety Inventory, Quick Inventory of Depressive Symptoms and Social Value Orientation (SVO) Slider Measure[23] in a pen and paper questionnaire format. After these, participants completed two rating tasks in which they were asked to rate various (i.e. in total 100 unfair, fair and advantageous offers) Ultimatum offers coming from an anonymous proposer on a 1-9 Likert scale (dislike vs liking). All offers involved splitting £10 between the proposer and the participant, and the offer amounts were presented in pence unit. After that participants rated their proposers' (i.e. the confederate's) pictures displaying different facial emotions, on a 1-9 Likert scale (i.e. from negative to neutral to positive, 9 different emotions in 6 iterations, a total of 54 ratings). These rating tasks were administered to establish participants' baseline preferences independently, and the ratings were later used to construct computational models accounting for decision-making processes in the UG. The order of stimuli in both of these rating tasks were randomised for each participant in order to prevent the induction of systematic biases in perception and decision-making in the subsequent stages of the experiment. Participants of a secondary control experiment (n = 25) completed these rating tasks after the main experiment.

After the rating tasks, participants completed the UG while undergoing pupillometry recording (n = 44). The task consisted of 6 blocks of 40 trials each. Where available, each participant was paired with one of 11 different confederates. The participants were told that they would be playing an interpersonal negotiation game against another participant recruited from general public, and in the game they would be interacting with the other person through internet connection to an online server. To strengthen the confederate manipulation, the participants were informed that their proposers attended on 2 occasions, and in their first visit they also respond to the same questionnaires as the participant did, and their pictures were taken while displaying different emotions, later to be used in the negotiation game. For some participants (47.7%) it was not possible to arrange a confederate due to feasibility issues (e.g. a mismatch between the participant, pupillometry testing room and confederate availability). Those individuals were explicitly told that they would be playing against a computerised strategy which was developed based on the behaviour of a previous participant. We also used this manipulation in our previous work to investigate value computations underlying human proposer behaviour in the UG[67]. In fact, all participants played against the same computerised strategy which was developed to sample offers and facial emotions probabilistically from two independent sliding windows.

In descriptive terms, the proposer strategy was designed to test the participants' acceptance threshold by sampling offers probabilistically around the threshold (e.g. previously rejected offers can stay the same mimicking an insisting behaviour, improve or even get lower), and displays negative facial emotions with relatively higher probability when the offers are rejected or displays positive facial emotions with relatively higher probability when the offers are accepted (details of the computer strategy is available in Supplementary Materials). We used a sliding window approach to make sure that both facial emotion stimuli and offer amount changes in small increments determined by these transition probabilities, which would allow us to present a gradually changing social interactive decision-making environment to the participants. A full debriefing letter summarising the aims and objectives of the study along with reasons for deception was provided at the end of the study.

At the beginning of the UG experiment, the participants were explicitly told that their proposers would be selecting one offer out of a window of different offers to make a proposal. Participants were told that this is to make sure that their proposers could not consistently make unfair or fair (i.e. 50/50 split) offers in which case the negotiation would get stuck in a limited range of offers. This measure was taken to make sure that the decision-making process was confined to responding to combinations of faces/offers in a gradually evolving task environment, and the influence of higher-order cognitive processes (e.g. Theory of Mind tracking[46], learning about the proposer's strategy) is limited. We think that UG is a particularly suitable task for this purpose as it allows reducing model complexity (e.g. eliminating recursive models), relative to other tasks such as the Stag Hunt[47], the Trust[68] or the Inspection games[45]. This is because the some economic decision-making models[5] posits that any gain is better than no gain and all offers should be accepted. Secondly, recursive ToM models with which participants can try to influence the proposer behaviour would only work effectively if the proposer's subsequent offer is more than twice as good as the offer rejected on the current trial, as otherwise the reward rate per trial cannot exceed the reward rate per trial if all offers are accepted. However, we minimised this possibility in our task design (~9% of trials, offer amount (mean ± SD): 132.6 ± 38), as offers were drawn from a sliding window and did not increase in a multiplicative manner. In the secondary/control experiment, we wanted to eliminate the effects of task instructions or the effect of the time point in which the rating tasks were completed (pre or post main experiment) and still observed comparable results to the original study. This seems to suggest that our initial approach with detailed task instructions might have been overcautious and rather unnecessary.

Just like in any traditional UG experiment, participants were told that their task is to accept or reject these offers coming from the proposer. The participants were told that the accepted offers will be distributed as proposed, but if they reject an offer both sides would get nothing for that trial. The participants were told that after their decision, the proposer would see 9 faces displaying different emotions (i.e. the same 9 faces the participant rated previously) and would select one to communicate how s/he feels in response to the participant's response and in the next trial the proposer would see another set of options, also giving him/her an opportunity to revise his/her offer. The participants were told that each block would start with a neutral face and a fair offer (i.e. 50/50 split) and can go any direction from that point onwards based on their negotiation choices. Within each trial the key epochs were: proposer's facial emotion, the offer, decision input from the participant, monetary outcome and proposer's emotional reaction to the participant's decision (Fig. 1). To establish continuity in the game, the proposer's emotional reaction at the end of trial t-1 would be the first stimuli presented on trial t.

The participants were instructed that at the end of the game a computer algorithm would randomly select 20 trials and the outcome of those trials would be paid to each side. These 20 trials used for compensation were selected irrespective of participants' decisions (i.e. regardless of whether participant had accepted or rejected that trial's offer). Participants were told that they may be inclined to accept all offers so that those randomly selected 20 trials would always have a monetary outcome for the participant, but in that case, we told them, that their proposer may detect this tendency and try to make offers as low as possible. We then told

participants to use the accept/reject responses strategically to negotiate better terms for themselves. These study design decisions allowed us to address our primary research questions in an unbiased manner: (i) the degree to which proposer's affective reactions influence participant choice behaviour; (ii) how this affective information is incorporated in decision values in iterative games.

During pupillometry recording, participants' heads were stabilised using a head-and-chin rest placed 70 cm from the screen on which an eye-tracking system was mounted (Eyelink 1000 Plus; SR Research). The eye-tracking device was configured to record the coordinates of both of the eyes and pupil area at a rate of 500 Hz. The pupillometry data collection lasted ~70 min per participant.

**Modelling biases in facial emotion recognition**. We modelled participant rating responses to affective faces (on a 1-9 Likert scale from negative to positive) by exploiting the properties of the 2-parameter probability weighting function[69]:

$$\varepsilon = e^{(-\delta(-\ln(\varphi))^{\gamma})} \tag{1}$$

where $\varphi$ is the true emotional state of the proposer's face as displayed and can take values from 0.1 to 0.9. The parameters $[\delta, \gamma]$ determine the curvature of the weighting function and where it crossed the $\varepsilon = \varphi$ diagonal line (Fig. 2a). We implemented the probability weighting function like a nonlinear regression model that minimises the difference between model estimates and participant's ratings (after performing a simple linear transformation by dividing the Likert ratings by 10, such that both the ratings and the estimates are bounded by 0 and 1).

**Modelling liking ratings for Ultimatum offers**. In line with previous literature[63], we modelled participants liking ratings ($\chi$) for each Ultimatum Offer (again, based on participant ratings on a 1-9 Likert scale) with a multiple linear regression model.

$$\chi = \phi_0 + \phi_1 R_S + \phi_2 |R_S - R_O| \tag{2}$$

where $R_S$ is the self-reward and $R_O$ is the reward amount to the proposer, and absolute value difference accounts for how much the participant cares for inequality. Here, it is important to clarify that because all offers involve splitting £10, $R_O$ is not entered into this regression model as an independent regressor as all monetary information is conveyed by the self-reward amount and the inequality term.

**Modelling participant choice behaviour in the Ultimatum Game**. The model-free analyses of participant choice behaviour indicated that both offer amount and the proposer's facial emotions should influence how people generate decision-values in the UG. Here, we formally describe all the models that we considered, to identify the cognitive model which accounts for participant choice behaviour the best. All models were fit to participant choices individually and 2nd level analyses were performed on the parameter estimates from the best-fitting model which is identified by a well-established Bayesian model selection approach[37]. In these behavioural models, we used task components from the current trial instead of modelling the complete task history as an approximation that is commonly performed in tasks in which the information in the current trial is dependent on outcomes in the preceding trials.

The first model that we considered assumes that participants construct the decision-value ($\tilde{v}$) of each offer utilising a Constant Elasticity of Substitution (CES) function, commonly used to account for consumer behaviour[70,71]. According to the CES function, the decision-value is computed as:

$$\tilde{v} = (\alpha R_S^{\rho} + (1 - \alpha)|R_S - R_O|^{\rho})^{1/\rho} \tag{3}$$

where $\rho$ is a nonlinear power utility parameter determining the concavity of preferences and $\alpha$ determines how much weight is assigned for the self-reward magnitude or the absolute value difference between the self and the other's reward magnitude (i.e. the inequality term).

We considered another model in which the decision-value is generated by comparing the inequality term relative to one's fairness threshold:

$$\tilde{v} = \lambda - |R_S - R_O| \tag{4}$$

where $\lambda$ is the fairness threshold parameter freely estimated between 50 and 950 (i.e. bounded by the minimum and maximum offer amount), defining the participant's subjective threshold in term of what they regard as acceptable.

The next model assumed that participants accept or reject offers based on their liking ratings, as established by the independent rating task participants completed prior to the social interactive decision-making task. This model allowed us to validate previous accounts of value-based decision-making which demonstrated that choice preference does not always align with participant liking ratings[72]. In this model, participant's liking of offers during the UG is estimated by feeding each participant's estimated coefficients from the linear regression model back to Eq. (2). Here, the decision-value is equal to:

$$\tilde{v} = \chi \tag{5}$$

We also considered another model with a similar structure in which the decision-values are generated online during the social decision-making task, instead of depending on participant's previous liking ratings. This model addresses the prediction that there will be a dissociation between how much people like offers

when these ratings do not have any financial consequence and how they value them in a social interactive context with monetary consequences (i.e. the fact that participants would be paid the outcome of 20 randomly selected trials, including both accept and reject decisions). Here, the subjective decision-value for each participant is computed by the following formula:

$$\tilde{v} = \kappa R_S - \omega |R_S - R_O| \tag{6}$$

where $\kappa$ and $\omega$ are free parameters modulating self-reward amount and the inequality terms, respectively. For model simplicity we used a single inequality aversion parameter, instead modelling advantageous and disadvantageous inequality separately. All models assume that conditions with relatively higher subjective value should be more likely to be accepted and participants' acceptance probability is generated by a sigmoid function:

$$q_A = 1/(1 + e^{-\beta\tilde{v}}) \tag{7}$$

where $\beta$ is the inverse temperature term which modulates the stochasticity of participant choices.

We first chose between these models which define participants' likelihood of accepting an offer solely based on the numerical components of the offer amount. We implemented this reduction in model space based on our model-free analysis which suggested that offer amount has a greater influence on acceptance probabilities than the proposer's facial emotions. Group-wise sum of Bayesian Information Criterion (BIC) scores indicated that the model described by Eqs. (6) and (7) was the best-fitting [inequality aversion] model to account for how offer amounts were translated to decision-values.

We then used this best-fitting model from the first stage as a template to further evaluate the degree to which facial emotions of the proposers dynamically influence the way decision-values are computed in the social decision-making task on a trial-by-trial basis. Based on previous literature, we considered a model in which the effect of proposer's facial emotion depends on the participants' individual variability in how malleable they are to external emotional influence[73]:

$$\tilde{v} = \kappa^{(\varepsilon - \mu)} R_S - (\omega^{(\varepsilon - \mu)} |R_S - R_O|) \tag{8}$$

where parameters $\kappa$ and $\omega$ are estimated between 0 and 10 and determine the degree to which the modulations of self-reward and the inequality term are subject to emotional influence. In this model, perceived emotions (Eq. (1)) act like an exponential function to modulate this influence parameter [6] and $\mu$ is the value where perceptual biases crossover the diagonal line (Fig. 2a, $\mu = 0.4$). We considered 3 variants of this model where proposer's facial emotions influence the self-reward amount ($R_S$) or the inequality term ($|R_S - R_O|$), or both independently.

We also investigated whether the proposer's facial emotions influence the inequality term in a parabolic, rather than an exponential functional form:

$$\tilde{v} = (\kappa + (\varepsilon - \mu)^2) R_S + (\omega + (\varepsilon - \mu)^2) |R_S - R_O| \tag{9}$$

For any given offer amount, this formulation relaxes the negative modulation of inequality as the proposer's facial emotions are gradually getting negative or positive, while assuming that neutral faces would have limited influence on the inequality term. As for the Eq. (8), we considered 3 variants of this model as well, facial expressions modulating either the self-reward magnitudes, or the inequality term or both.

Finally, we considered a competing model which assumes that the proposer's facial emotions influence the value of offers through a weighted integration.

$$\tilde{v} = w\tilde{v} + (1 - w)(\varepsilon - \mu) \tag{10}$$

Here, $w$ is a free parameter estimated between 0 and 1, and determines the degree to which the participant assigns credit to subjective value of the offers and/or to the proposer's facial emotions which are represented nonlinearly. The additive integration means that other's facial emotions should have an intrinsic value, an assumption that is in line with existing literature[74].

All model parameter estimation followed our existing protocols[26,28,50], namely a Bayesian model fitting procedure, by calculating the full joint posterior distribution of the parameters over the whole parameter space, and deriving exact parameter values by integrating these probabilities with their corresponding discrete parameter values. A Bayesian model selection method was implemented to choose between these 11 models.

**Pupillometry data preprocessing**. Eye blinks were identified using the built-in filter of the Eyelink system and were then removed from the data. A linear interpolation was implemented for all missing data points (including blinks). The resulting trace was subjected to a low pass Butterworth filter (cut-off of 3.75 Hz) and then z transformed across the session[26,27]. The pupil response was extracted from each trial, using a time window based on the presentation of the offer amount. This included a 7.5 s baseline period before the presentation of the outcome (including the period at the beginning of a trial marked by a fixation cross and presentation of the proposer faces alone, Fig. 1), and a 4.5 s period following offer presentation. Baseline correction was performed by subtracting the mean pupil size during the 7.5 s baseline period prior to the presentation of each offer, from each time point in the decision and outcome period[75]. This baseline correction, which also included the first facial expression shown within a trial (see Fig. 1), allowed us to extract the phasic pupillary response and controls for potential fluctuations in luminosity within each trial.

Individual trials were excluded from the pupillometry analysis if more than 50% of the data from the outcome period had been interpolated (mean = 10.9% of trials)[26]. The preprocessing resulted in a single sets of pupil time-series per participant containing pupil dilation data for each of the included trials.

**Regression analysis of pupillometry data**. We implemented an OLS model in a similar manner to a functional neuroimaging (fMRI) analysis, by fitting the regression model to the pre-processed pupillary data at every 2-ms time point. The pupil model had 11 regressors as defined by the following linear regression equation:

$$pupil(2\pi r) = \phi_0 + \phi_1 i + \phi_2 \kappa R_S^i + \phi_3 (\omega + (\varepsilon - \mu)^2)^i + \phi_4 |R_S - R_O|^i + \phi_5 surprise_{offer}^i$$
$$+ \phi_6 surprise_{faces}^i + \phi_7 vol_{offer}^i + \phi_8 vol_{faces}^i + \phi_9 noise_{offer}^i + \phi_{10} noise_{faces}^i$$

$$(11)$$

Here we provide brief descriptions for each regressor in the model. The regressor for the constant term ($\varphi_0$) captures the variance in pupil size across all trials in the task. The regressor ($\varphi_1$) accounts for the effect of trial numbers, as a proxy for accumulated fatigue ($i$: taking values from 1 to 240). We entered one regressor ($\varphi_2$) which represents the subjective value of self-rewards under the best-fitting behavioural model. The regressor ($\varphi_3$) encodes how the perceived facial emotions modulate the subjective inequality parameter (Eq. (1)). The regressor ($\varphi_4$) encodes the magnitude of the self–other inequality. The rest of the regressors were defined by a recursive Bayesian filter that we recently reported[27] which can optimally track the hidden structures (i.e. environmental volatility, *vol* in the above formula, and noisy fluctuations in the environment) of dynamically changing environments. The surprise regressor was congruent to the conditional –log probability of an offer coming from a distribution with mean and standard deviation (SD) estimated by the Bayesian filter. This would mean that stimuli which violate the expectations of the observer should lead to a greater pupillary surprise signal. This regressor was calculated for both the offers and the proposer's facial emotions. All regressors were demeaned prior to model fitting. Fitting a multiple linear regression model to pupillary data is akin to analysis of fMRI datasets and allows regressors to compete for variance in the data. Therefore, resulting regression coefficients no longer need to be subjected to further correction for multiple comparisons by the number of regressors in the model.

**Statistics and reproducibility**. We used appropriate linear and logistic regression models as described in individual sections of this manuscript. Where applicable, the main effects were tested by *t*-tests from baseline and follow-up tests were corrected for multiple comparisons. We confirmed the reproducibility of the main experiment by a control experiment conducted on an independent cohort ($n = 25$, details below).

**Control experiment**. In order to rule out the possibility that the model-free effects that we demonstrated in Fig. 2 and model-based results shown in Supplementary Fig. 3 were an artefact of detailed task instructions and/or the order of rating tasks (i.e. pre-versus post-experiment), we collected data from an independent sample of participants ($n = 25$, Supplementary Table 1 for demographic information). Effect sizes estimated from the main experiment indicated that a sample size larger than 20 is more than adequate to probe both components of the decision problem (i.e.. facial emotions and offer amount). Effect sizes for these main task components in the control experiment were $d = 2.626$ for the facial emotions and 1.833 for the offer amount. Based on the finding that different confederates did not have any influence on behavioural results and there were no differences between human and computerised opponents (Fig. 5), participants in the control experiment performed the task explicitly knowing that they will be interacting with a computerised strategy that is designed to mimic human proposer behaviour. This was also a necessity for the control experiment as COVID-19 restrictions in the UK did not allow introducing people who are unknown to each other solely for experimental purposes. In this cohort, the participants were explicitly instructed to make decisions freely as they liked and they completed both of the rating tasks after completing the main social interactive decision-making experiment. Both the model-free (Supplementary Fig. 8) and model-based (Supplementary Fig. 9) results were comparable with those reported from the main experiment.

**Reporting summary**. Further information on research design is available in the Nature Research Reporting Summary linked to this article.

## Data availability
All the raw data from participants is deposited on the servers of the Open Science Framework: https://osf.io/fjvrh/. Any other data related enquiry can be addressed to the corresponding author.

## Code availability
All the key analysis and model scripts are deposited on the servers of the Open Science Framework: https://osf.io/fjvrh/. Any other code related enquiry, including the experimental task can be addressed to the corresponding author.

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

## Acknowledgements

This study was funded by National Institutes of Health Research (NIHR) Oxford Biomedical Research Centre funding allocated to E.P. and University of Oxford Department of Psychiatry funding allocated to E.P. and J.X. The views expressed in the manuscript are those of the authors and not necessarily those of the NHS, the NIHR or the Department of Health. M.B. is supported by a Clinician Scientist Fellowship from the MRC (MR/N008103/1) and by the NIHR Oxford Health Biomedical Research Centre. E.P. and C.J.H. are supported by a joint funding from the UK Medical Research Council and Janssen Pharmaceuticals (MR/S035591/1) awarded to C.J.H. The authors would like to thank Dr. Alexander Kaltenboeck for his help with the study.

## Author contributions

E.P. designed the study. D.A.J.M., J.X. and E.P. collected the data. E.P. analysed the data. M.B. developed the recursive Bayesian filter. D.A.J.M., J.X., C.J.H., M.B. and E.P. contributed to writing up of the manuscript and to the critical revisions in subsequent stages.

## Competing interests

E.P. has received consultancy fees from Janssen Pharmaceuticals. M.B. has received travel expenses from Lundbeck for attending conferences and acted as a consultant for Jansen Research and CHDR. C.J.H. has received consultancy fees from p1vital. Lundbeck, Pfizer, Sage Pharmaceuticals, Servier, Zogenixs and J&J. D.A.J.M. and J.X. do not have any competing interests.
