## [Peer Review File · Communications Biology]

Reviewers' comments:

Reviewer #1 (Remarks to the Author):

General

In this paper, Murphy et al. examined the effect of a proposer's facial emotions on a receiver likelihood to accept offers in an interactive ultimatum game (UG). They demonstrate that facial emotions affected participants' inequality aversion. I think that this finding is interesting and novel, but I have some comments and concerns. Especially, the task is not clearly described, and it is hard to understand the findings and different analyses, and to interpret them.

Importantly, the manuscript is very hard to follow, including some redundant information, while other parts are missing, and other scattered in different parts.

They also showed that pupil diameter was related to the offer's inequality. While interesting, this finding seems disconnected from the main theme of the paper.

While I think that the study is interesting and includes clever modelling, I have a number of comments and questions, detailed below.

1. Theoretical framework and hypotheses

The authors provide very little theoretical basis for the effect of emotional face expressions on decisions in games. There is a substantial literature on this topic, and they cite some findings, but without addressing the relevance of player's emotions to games. For example, one may be motivated to make other players happy, regardless of monetary outcome of the game, or facial expressions may signal intention, or have consequence in future interactions. This is completely absent from introduction and discussion, and it makes it hard to put these findings in context.

This is also the case when discussing pupil size. There is a generic and technical background on NE, pupil size and arousal, but nothing at all about any relevant predictions to the current study.

2. Task

The description of the task is scattered in different parts of the manuscript and very hard to follow. I am not sure that the UG is a zero-sum game, as there is at least one outcome where everyone loses – rejection of the proposal – and in all other cases both gain.

I am not sure that the current task is a UG at all. The participants are told that there is no risk in rejection, as these trials are not counted for their payment.

The participants got a lot of information about the task, with instructions on preferred strategies. Isn't this a confound?

In the task a face is presented between the participant's decision and the new proposal. This makes it very hard to understand what the facial expression is linked to – is the happy face the outcome of the participants' previous action, or associated with the new proposal (i.e. I am happy with this new proposal, or I am grudgingly propose this). These are very different interpretations of the task.

The authors instructed participants that the proposer chooses between a limited set of options.

This may put a lot of emphasis on facial expression – for example a sad face may indicate that the proposer knows it is not a good offer, but this may be the best of what he got. A participant may therefore be willing to accept to make the proposer happy, even though he does not like the offer.

The authors mention that the confederates were not instructed about emotions, but simply to make positive or negative face. How much variation was there in the interpretation of the confederates? Is it possible to see the faces? Angry/sad/disgust are all negative but may signal different social information.

I understand that the task included two processes – the proposed offer 'staircased', i.e. more generous when rejected and less generous when accepted, and faces followed previous accept/reject decisions. This leads to interaction between processes, as indicated in supplementary figure 2 – there are areas of the face/offer subspace that are seldom sampled. Does this pose a problem on interpreting the relations between these parameters?

Also, this dual process does not seem to be modeled – there is no model that includes both previous decision and current facial expression.

And again – do happy faces signal an outcome of previous offer or information about the new offer?

3. Results

I found the asymmetry in figure 3A very interesting – it looks like participants are willing to accept an offer that is too generous from a positive emotion face (bottom right) but not from a negative emotion face (top right). In the left side of the matrix, with offers that are too small, this

asymmetry seems to disappear, and they are in general likely to reject. To me it seems to suggest that facial expressions are linked to the upcoming offer, and indicate how happy the proposer with the offer. A sad and too generous proposer may lead to rejection, as the layer feel that he is taking advantage of the proposer. A happy and too generous proposer may indicate that he is happy with this, encouraging the participant to take the offer. Such perspective of signaling is not considered at all in the models, theory and background.

This asymmetry between stingy and over-generous offers is completely ignored, as both are modeled as 'inequality' offers (absolute value). Isn't this an important aspect of the behavioural results?

Machines vs. humans – previous works investigated learning about generosity of humans and machines (Hackel et al, Haggard et al and others). I find it surprising that there was no effect here, but it could be the small number of participants.

I am concerned that the rating of all faces and offers before the task interfered with the reliability of the social interaction, or at least with the effect of faces in the game. This may also explain why there was no difference between interaction with humans and algorithms.

The use of SVO seems circular. SVO measures exactly the same thing as the main task – splitting sums of money with another person (albeit not exactly UG as the sum of money for self and other may increase). It is therefore encouraging that both measures agree, but I don't think that anything new could be inferred from this result.

The logistic regression looking at the effects of previous interactions on current choice does not seem to include the current proposal and facial expression.

4. Model

The model description is scattered in many places, and the short description in the results section does not give enough information about the model and its contribution to understanding the task. The model very cleverly uses the individual's subjective evaluation of facial expressions. However, it does not seem to incorporate any aspect of the interactive nature of the task (the two sliding windows described in the task section, the effect of previous choices). As such it seems to capture the responses as if they are independent from one another, which is exactly what the authors state that they try to avoid and overcome in the introduction. What does it reveal beyond the interaction effect observed in the standard analysis?

5. Pupils

The results seem interesting, just completely unrelated to any other part of the results.

6. Structure

This paper was not easy to follow. It feels like it was originally written in a 'methods-first' format and hastily transformed to 'methods-last' format. The reader is constantly surprised by unexplained bits of information (the algorithmic vs human proposer is mentioned in one time), new questionnaires, tasks and ratings are presented with no clear indication to their order or any related hypothesis. The demographics are part of the results, presented first, and not a part of the participants section in the methods.

Some parts of this manuscript seem disconnected from one another. The generic introduction about pupils and NE, and indeed the entire pupil diameter analysis, do not integrate with the rest of the paper. The long discussion regarding FERT (with no mention of what this acronym represents), presentation of stimuli, response time, anxiety and depression seems more suitable in other parts of the manuscript (methods? Supplementary materials?).

The introduction starts with bargaining example of head of state, but it seems weakly connected to the problem at hand (no emotions or facial expressions are mentioned in the example, and there is a lot of literature about intergroup relations and bargaining). Also, stating that there is no information about the computational and physiological mechanisms of iterative social interactions is too broad and wrong – the next paragraph considers such examples, and there are lots of computational models of iterative games (game theory perspective, decision making perspective POMDP), as well as behavioural data (iterative prisoner dilemma, iterative trust games, iterative stag hunt) and neurological data (Autism and UG by Montague for example).

Overall, I think that a lot of editing is needed in order to piece together all the scattered information, to edit out irrelevant information, and to add relevant background works.

Reviewer #2 (Remarks to the Author):

This is my first review of the manuscript COMMSBIO-21-0206 entitled "Dynamic modulation of inequality aversion in human interpersonal negotiations".

Summary

In this paper, the authors use a repeated version of the Ultimatum game to investigate human negotiation in gradually evolving environments. In this study, the proposer's facial emotions are expressed and communicated to the receiver before the offer is made and after the respondent decided whether to accept or reject it. The authors propose and test different models to describe possible drivers of the observed human choice behavior. They found that both the proposer's facial emotions and the amount offered affect the choice behavior of the respondent. In a subsequent model-based analysis the authors found that the decisions of the respondents are guided by aversion to inequality. They found that the best-fitting model is the one that assumes non-linearity in the way the proposer's facial emotions are perceived and act on participants' perception of inequality. Finally, the authors recorded pupil dilation and found that pupil size encodes the magnitude of self – other inequality prior to decision onset.

General evaluation

The manuscript is written clearly and the research questions addressed in this study are particularly relevant. In particular, the idea to test how trial-by-trial computation of decision values underlying human responder behavior takes place when information about the affective state of the proposer is available before and after the respondent decision is an interesting and challenging research question. Data analyses are conducted in a logical and rigorous manner, and there are many interesting results reported in the paper. The only main concern and limitation of this research are in the experimental procedure implemented by the authors. The description of the experimental procedure is not particularly clear and leaves several doubts on the real possibility of considering the observed choices of the respondents as the result of natural social interaction. In particular, I am wondering if the experimental procedure proposed by the authors may have induced the respondents to make choices that are not in line with their real strategic preferences. I will try to be more precise in what follows:

- At the beginning of the experiment, participants were told that their proposers would be selecting one offer out of a window of different offers. It is unclear what the participant knows about this window. Does the participant receive information about how the window is selected and the size of the window? From what I understand, by reading the procedure, the participants did not receive any information about this window. In my opinion, this procedure determines different sources of uncertainty for the participants: Is the offer low or high because of the (randomly???) selected window? What is the proposer trying to tell me with this emotional expression? Maybe that he could not offer a higher amount? The authors introduce a source of uncertainty in the respondent decision. If the respondents are unaware of the possible offers available to the proposer, then the participants are not playing a repeated Ultimatum game but a more complex game with private information.
- The authors state that this measure (an offer selected out of a window of different possible offers) was taken to be sure that the decision-making process was confined to responding to combinations of faces/offers in a gradually evolving task environment and the influence of higher-order cognitive processes (theory of mind, learning about the proposer's strategy) is limited or become redundant. Personally, I cannot see why increasing the complexity of the strategic interaction can limit theory of mind processes or reduce the respondent's propensity to infer the rationale behind the decisions of the proposer.
- The authors told participants that "they may be inclined to accept all offers so that those randomly selected 20 trials would always have a monetary outcome for the participant, but in that case, we told them, that their proposer may detect this tendency and try to make offers as low as possible". Then they told participants "to use the accept/reject responses strategically to negotiate better terms for themselves." In my opinion, this is a clear demand effect. This suggestion clearly may have prompted the participants to choose differently from their initial strategic preferences. The observed behavior might be significantly different to the one we would have observed if participants had been free to use their own preferred strategy. This rise some doubt about the generalizability of one of the main result of the paper: the best fitting model, that predict 86% of participants decisions in a repeated Prisoner's Dilemma, is the one that assumes non-linearity in the way the proposer's facial emotions are perceived and act on participants' perception of

inequality. We do not know if this model would still be the best fitting model of the natural behavior of the respondents. Similarly, this suggestion might have affected the results of the logistic regression model: "stimuli shown in proceeding trials as well as participants' previous decisions influenced their choice behavior on the current trial". On one side, the authors are trying to reduce the respondent's propensity to infer the rationale behind the decisions of the proposer, on the other, they told participants to be strategic and use accept/reject responses to negotiate better terms for them. If one of the main objectives of the paper is to increase the "ecological validity in terms of capturing real-world human social interactive decision making processes" this cannot be achieved "driving" the behavior of the participants.

Other comments

- Lines 102 - 104: provide a description, somewhere in the paper, about how these sliding windows with transitions probabilities work.
- Lines 287 - 292: The opponent-type regressor which determines whether the participants played against a human or a computerised opponent did not have any significant main effect on human behaviour or parameter estimates. It is possible that recent advances in AI mimicking, and even excelling, human behavior in competitive games might have an indirect effect on these results. An alternative hypothesis is that the choice behavior of the participants is in both cases (human vs computerized opponent) simply following the "suggestions" provided by the experimenters about the decision strategy that they should implement. This would reduce significantly possible differences in the two conditions.
- Lines 457 - 464: the authors state that there is no work prior to the present study that has on pharmacological manipulation of the central NE system and how this would affect perceived inequality in humans. This part is confusing; in this study there is no pharmacological manipulation.
- In general, the discussion is too long and some parts are not strictly linked to this research work. In particular, the second part of the discussion can be significantly shortened.

Reviewer #3 (Remarks to the Author):

The manuscript by Murphy et al. describes how humans behave in a version of the ultimatum game which allows studying the interplay between the proposer's facial emotions and the offer amounts.

To review the paper, I focused on the main criteria displayed in the journal's website (<https://www.nature.com/commsbio/about/aims>). In my opinion, while this paper shows additional evidence that emotions affect economic choices, I do not believe that the results reported here represent "an advance in understanding which may influence thinking in the field" (including the sub-field of human economic decision making).

NOVELTY OF THE RESULTS

The novel contribution of the paper compared to the existing literature remains unclear. In the Introduction, the authors make several strong claims about novelty which are either overly specific (point 1) or too general to be true (point 2):

1) Lines 83-84: "how trial-by-trial computation of decision values underlying human responder behaviour takes place in an iterative UG with an ecologically valid affective component has not been shown before". There are several papers that have used computational approaches to analyse trial-by-trial variability in iterative UG and reported more than just "summary statistics of average acceptance probabilities" (line 82). For example, Gabay et al. (2018) used a statistical model to study the psychopharmacological effects of MDMA and psilocybin on responder behaviour in the UG at the trial-by-trial level (<https://www.nature.com/articles/s41598-018-26656-2>). Si et al. (2020) implemented a decoder model that successfully predicts responder behaviour using trial-by-trial variability in the EEG signal (<https://www.sciencedirect.com/science/article/pii/S1053811919309243>). Xiang et al. (2013) even used model-based fMRI and implemented an ideal-observer computational model to show how signatures of trial-by-trial computations of decision value are encoded in the striatum, insula, and OFC (<https://www.jneurosci.org/content/jneuro/33/3/1099.full.pdf>).

While none of these studies have an “ecologically valid affective component”, this can also be said to be the case in the present study (where do we set the threshold for “ecological validity”?). And even if one believes that this study has an ecologically valid affective component, then the main contribution of the paper would be simply extending a widely used approach to a more ecological setting, something that is unlikely to represent a significant advancement in the understanding of economic decision-making.

2) Lines 140-141: “the role of pupil-linked central arousal systems during social interactive decision-making remains unknown”. The claim here is simply too general to be true (social decision making includes, for example, the analysis of moral choices which have been studied with pupillometry). In fact, there is even one study that has linked pupil size with participants’ behaviour in the UG (Rago et al. 2015, <https://www.tandfonline.com/doi/abs/10.1080/17470919.2015.1057295>).

Overall, this left me unconvinced that the paper makes a contribution to the literature that is both meaningful and novel. Moreover, there are other papers that have shown evidence for the main conclusion of the paper (“inequality aversion is a malleable cognitive process”). For example, Vavra et al. (2018) showed how simply changing people’s expectations about what constitutes a standard offer modifies people’s behaviour in the UG. One could say that this paper already showed that aversion to inequality is a malleable process (<https://www.frontiersin.org/articles/10.3389/fpsyg.2018.00992/full#B24>)

EVIDENCE FOR THE CONCLUSIONS

1) The downside of breaking trial-wise independence. One of the main features of the paper is that the task uses an “ecologically valid” experimental design. However, participants know they are interacting with an avatar and so their behaviour might be completely different to the one observed in real pairwise interactions between individuals. In fact, one caveat of breaking inter-trial independence (to gain in ecological validity) is that the authors are also breaking the statistical independence between stimuli presented at different trials, which are used as different predictor variables in a multivariate regression. The paper does not provide any evidence that this approach is justified and that there is sufficient unshared variance between consecutive stimuli (what is the variance inflation factor of the multivariate regressions?)

2) No evidence for an effect vs. evidence for no effect. Figure 2B is supposed to convey the message that the model parameters fitted in Eq. 1 are not influenced by demographic factors and symptoms of depression/anxiety. The evidence presented for this conclusion is clearly insufficient. Showing that regression coefficients are within an interval including zero (i.e., showing that beta values are n.s.) might simply mean that the study is underpowered to detect those effects (something that could be the case given the large C.I.). In other words, the study shows no evidence for an effect, and the authors report this as evidence for a null effect.

TECHNICAL SOUNDNESS OF THE DATA

Line 111: “Our a priori hypothesis was...” With the data presented in this paper, the reader cannot be confident enough that the presented data tested “a priori” hypotheses since there is no pre-registration of the hypotheses, methods, or analyses. In fact, some passages of the paper suggest that results went in an unexpected direction. For example, the authors report a rather counter-intuitive non-linear relationship between actual and perceived emotion (line 170: “This bimodality indicated the possibility of a nonlinearity in human facial emotion recognition (...). To capture this nonlinearity,...”) This suggests that a main feature of the task (perceived emotion) follows a complex rule that was added ad-hoc to explain a pattern that was unexpected by the authors. While there is nothing inherently wrong with this situation (empirical science often goes in unexpected ways), the authors should be crystal clear about which of the hypotheses and analyses were exploratory and which of them were confirmatory. Moreover, if one or more of the presented models were selected in a post hoc data-driven manner, it would be reassuring for the reader that the results replicate in a second dataset (ideally pre-registered).

Reviewers' comments: **[responses to Reviewers are typed in bold]**

Reviewer #1 (Remarks to the Author):

General

In this paper, Murphy et al. examined the effect of a proposer's facial emotions on a receiver likelihood to accept offers in an interactive ultimatum game (UG). They demonstrate that facial emotions affected participants' inequality aversion. I think that this finding is interesting and novel, but I have some comments and concerns. Especially, the task is not clearly described, and it is hard to understand the findings and different analyses, and to interpret them. Importantly, the manuscript is very hard to follow, including some redundant information, while other parts are missing, and other scattered in different parts.

They also showed that pupil diameter was related to the offer's inequality. While interesting, this finding seems disconnected from the main theme of the paper.

While I think that the study is interesting and includes clever modelling, I have a number of comments and questions, detailed below.

We thank the reviewer for this set of feedback and a positive evaluation on our modelling approach. In the revised submission we endeavoured to make changes that can fully address the reviewer's recommendations.

1. Theoretical framework and hypotheses

The authors provide very little theoretical basis for the effect of emotional face expressions on decisions in games. There is a substantial literature on this topic, and they cite some findings, but without addressing the relevance of player's emotions to games. For example, one may be motivated to make other players happy, regardless of monetary outcome of the game, or facial expressions may signal intention, or have consequence in future interactions. This is completely absent from introduction and discussion, and it makes it hard to put these findings in context.

In the revised submission we endeavoured to include more of the literature that we might have overlooked in the original submission. Regarding the effect of facial emotions on decision-making, we have included a few more studies into the introduction and decided not to include some of the other studies which focused on the effect of emotions displayed by the responders (e.g. Reed et al, 2014 and Schreiner 2009). Particularly the literature that we came across mainly utilised one-shot ultimatum game designs and to the best of our knowledge the influence of affective faces on interactive decision-making is not mentioned. If the reviewer had specific papers in mind, we would be more than happy to make necessary revisions to include these in the next iteration.

This is also the case when discussing pupil size. There is a generic and technical background on NE, pupil size and arousal, but nothing at all about any relevant predictions to the current study.

Again, apart from a single study which was recommended by Reviewer 3, we did not come across any fundamental studies looking at decision-making in the UG and its pupillary correlates. The paper recommended by the Reviewer 3 is now included in the introduction although the research sample consisted of children and a one-shot task design [In 152]. If the reviewer had any specific

papers in mind, we would be more than happy to include them in the next iteration. Due to previous findings related to autonomic nervous system response to unfair offers, we predicted that pupil size may be sensitive to the magnitude of inequality which is included in the final paragraph of the introduction.

2. Task

The description of the task is scattered in different parts of the manuscript and very hard to follow.

I am not sure that the UG is a zero-sum game, as there is at least one outcome where everyone loses – rejection of the proposal – and in all other cases both gain.

We agree with the reviewer's point and revised the section of the paper to highlight that zero-sum classification applies only to offers that are accepted.

I am not sure that the current task is a UG at all. The participants are told that there is no risk in rejection, as these trials are not counted for their payment.

We would like to clarify that the random selection of trials at the end of the game also included those with rejection responses, such that if the participant rejected all offers on those randomly selected trials, they would not be paid any additional money. Therefore, we think that the experiment is in line with the main rules of the ultimatum game.

The participants got a lot of information about the task, with instructions on preferred strategies. Isn't this a confound?

We agree with the reviewer that we have given extensive information about the task to the participants due to the complexity of the experiment, to make sure that participants understood the task. In the revised submission, we conducted an additional control experiment to rule out the effect of task instructions or time point in which participants completed the rating tasks. The results of the control experiment agreed with the initial results indicate that these may not be the primary driver of the participant choice behaviour.

In the task a face is presented between the participant's decision and the new proposal. This makes it very hard to understand what the facial expression is linked to – is the happy face the outcome of the participants' previous action, or associated with the new proposal (i.e. I am happy with this new proposal, or I am grudgingly propose this). These are very different interpretations of the task.

The authors instructed participants that the proposer chooses between a limited set of options. This may put a lot of emphasis on facial expression – for example a sad face may indicate that the proposer knows it is not a good offer, but this may be the best of what he got. A participant may therefore be willing to accept to make the proposer happy, even though he does not like the offer.

Thank you very much for this feedback and asking for further clarity. We would like to emphasise that the proposer's facial emotions were always linked to participant's decisions apart from trial 1 which always started with a neutral face and a neutral (i.e. 50-50) offer. We added the information below to the methods section of the manuscript for clarity:

[In 646]

"Within each trial the key epochs were: proposer's facial emotion, the offer, decision input from the

participant, monetary outcome and proposer's emotional reaction to the participant's decision (Figure 1). To establish continuity in the game, the proposer's emotional reaction at the end of trial t-1 would be the first stimuli presented on trial t."

The authors mention that the confederates were not instructed about emotions, but simply to make positive or negative face. How much variation was there in the interpretation of the confederates? Is it possible to see the faces? Angry/sad/disgust are all negative but may signal different social information.

We completely agree with the reviewer and our idea here was to allow some flexibility in the proposer's facial emotions, so that different labs around the world using this approach can replicate our findings which we think would not be possible if the definitions of negative or positive facial emotions are too much constrained. This is also the reason why we wanted to include 11 different confederates so that our findings would not be linked to a specific confederate image. We were able to obtain consent from two of our confederates to share their pictures as supplementary materials and this is now included in the revised submission.

I understand that the task included two processes – the proposed offer 'staircased' ,i.e. more generous when rejected and less generous when accepted, and faces followed previous accept/reject decisions. This leads to interaction between processes, as indicated in supplementary figure 2 – there are areas of the face/offer subspace that are seldom sampled. Does this pose a problem on interpreting the relations between these parameters?

We would like to highlight that the transitions between the states were always probabilistic, such that rejection of offers did not automatically need to more generous offers in the following trial. We agree with the reviewer that there are sections of the stimuli space which were sampled less than the other places. In fact, and for completeness we have shown model misprediction probabilities (Supplementary Figure 4) and descriptively this is not the mirror opposite of the number of trials sampled from each state space, indicating that lower number of trials sampled from a specific stimuli combination did not lead to poorer model prediction accuracy. Additionally, by running the experiment for a reasonably long time (240 trials for each participant) and in total 69 participants (the original and control experiment combined) we think we were able to capture the decision-making process as it would emerge naturally within the task environment. Finally, we think the stability of the model parameters suggests that they capture their specific domains reasonably well (Supplementary Figure 5).

Also, this dual process does not seem to be modeled – there is no model that includes both previous decision and current facial expression.

We would like to thank the reviewer for pointing out to this possibility, which we might have overlooked in the initial submission. We now included this analysis in our logistic regression analysis of the participant choice behaviour, which is reported in Figure 4. The updated results seem to suggest that there is no significant contribution from the current face, but there is a significant interaction between the current face and the previous offer, albeit at a lower degree of

significance relative to the facial emotion and the offer amount from the preceding trial.

And again – do happy faces signal an outcome of previous offer or information about the new offer?

Just to clarify a happier post-decision face from the beginning of the trial would signal the proposer's emotional reaction to participant's decision.

3. Results

I found the asymmetry in figure 3A very interesting – it looks like participants are willing to accept an offer that is too generous from a positive emotion face (bottom right) but not from a negative emotion face (top right). In the left side of the matrix, with offers that are too small, this asymmetry seems to disappear, and they are in general likely to reject. To me it seems to suggest that facial expressions are linked to the upcoming offer, and indicate how happy the proposer with the offer. A sad and too generous proposer may lead to rejection, as the layer feel that he is taking advantage of the proposer. A happy and too generous proposer may indicate that he is happy with this, encouraging the participant to take the offer. Such perspective of signaling is not considered at all in the models, theory and background.

We would like to thank the reviewer for his/her feedback on Figure 3A. We also think that, as far as we are concerned this is the 1st study to describe the ultimatum game decision-making behaviour in this manner. However, it is important to highlight that the heat map collapses all information across the trials, for example the same stimuli combination (e.g. facial emotion 6 x offer amount 350) might have occurred on trials let's say 5, 35, 111, 179 etc. Although these trials may have different meanings in an iterative context, the heat map would collapse across all to give an overall feeling of the decision probability for that combination. In that respect, it is more informative in conjunction with Figure 3B, just to confirm that the task effectively probes both the contributions of facial emotions and the offer amounts simultaneously. Consequently, we think the interpretation of facial emotions being linked to subsequent offer may not be accurate, because they change or stay the same probabilistically with respect to participant choice on each trial.

This asymmetry between stingy and over-generous offers is completely ignored, as both are modeled as 'inequality' offers (absolute value). Isn't this an important aspect of the behavioural results?

We agree with the reviewer that we did not model advantageous versus disadvantageous inequality separately. The reason for this decision is, doing so would significantly increase the number of models that we need to explore as each model can be split into 2 more models (parameter for advantageous versus disadvantageous inequality) and further interaction of these parameters with the facial emotions and we thought this might lead to model over-fitting (searching the best fitting model across 44+ models). We can also evaluate whether this would have a substantial benefit, as are currently proposed model predicts participant choice behaviour approximately at 86% and does not perform systematically poorly in one specific part of the stimuli space and it is able to regenerate the essence of original raw data. If the reviewer is keen to see this data, we would be happy to explore whether splitting the best fitting model further would improve the model predictive accuracy.

Machines vs. humans – previous works investigated learning about generosity of humans and machines (Hackel et al, Haggard et al and others). I find it surprising that there was no effect here, but it could be the small number of participants.

We also think this is an interesting feature of the data. In the revised submission, we provided a detailed discussion of Hackel et al paper recommended by the Reviewer [In 327], however we failed to identify Haggard et al paper on [social] reinforcement learning, but we would be more than happy to include this if the reviewer can share the exact reference. The discussion of Hackel et al highlight that human participants use trait generosity related models to learn about human opponents, when opponents are represented by abstract fractals. In fact, in our previous work where human participants acting as proposers were trying to learn about responders' acceptance tendencies, they were using SVO-related learning models, similar to what is being shown by Hackel et al. In that previous work, we had told that the computer strategy acting as the responder was modelled on the basis of human behaviour captured in an earlier study. This seems to suggest that, experimental manipulations like confederate designs or machines mimicking human behaviour (for example this element was not present in Hackel et al's work), may be probing human decision-makers to make choices similarly between human and computerised opponents.

In our additional control experiment (n=25) in which participants were explicitly told that they would be interacting with the computerised strategy modelled on the behaviour of previous participants and we have seen virtually similar behaviour to our original experiment. Although these sample sizes may look small, the effect sizes for the contribution of facial emotions and the offer amount are fairly large and this is what we wanted to capture in our current experiment. We think, with the incredible advance of machine learning and artificial intelligence (e.g. DeepMind's AlphaGo), human participants probably find it easier to attribute human features to computerised strategies than research done earlier. This is our descriptive explanation (and current hypothesis) of the lack of opponent type effects (although we now acknowledged the possibility that maybe our sample size was not large enough to detect such differences).

I am concerned that the rating of all faces and offers before the task interfered with the reliability of the social interaction, or at least with the effect of faces in the game. This may also explain why there was no difference between interaction with humans and algorithms.

Thank you very much for pointing out this possibility. In order to address that we recruited an independent cohort of participants who completed the same task but rated both the offers and the faces after the main experiment. The findings from this cohort are reported in Supplementary Figure 8 and reveals overlapping findings to our original experiment in terms of a significant contribution from facial emotions and offer amounts and our proposed model fitting still performs the best to the new data among all the models that we have considered.

The use of SVO seems circular. SVO measures exactly the same thing as the main task – splitting sums of money with another person (albeit not exactly UG as the sum of money for self and other may increase). It is therefore encouraging that both measures agree, but I don't think that anything new

could be inferred from this result.

We would like to thank the reviewer for this feedback. We wanted to keep the SVO measure in the study because although they concern similar inequality aversion processes, the slider measure shows a string of offers that the participant can choose from, whereas the model parameter is estimated from a binary decision. It is rather reassuring that the essence of the model parameter correlates with an external questionnaire measure, and we also wanted to keep this in to allow continuity with our previous work in which we investigated the proposer behaviour in ultimatum game (Pulcu and Haruno, 2020 Journal of Experimental Psychology Gen).

The logistic regression looking at the effects of previous interactions on current choice does not seem to include the current proposal and facial expression.

As briefly described in one of our responses above, we now included this analysis and reported in Figure 4.

4. Model

The model description is scattered in many places, and the short description in the results section does not give enough information about the model and its contribution to understanding the task.

The model very cleverly uses the individual's subjective evaluation of facial expressions. However, it does not seem to incorporate any aspect of the interactive nature of the task (the two sliding windows described in the task section, the effect of previous choices). As such it seems to capture the responses as if they are independent from one another, which is exactly what the authors state that they try to avoid and overcome in the introduction. What does it reveal beyond the interaction effect observed in the standard analysis?

We would like to thank the reviewer for the positive feedback about our modelling approach and these additional recommendations. One part of our response to these comments overlap with the explanation of the heat map results which summarise the model-free analysis of the data. This heat map collapses all the trials in which the same stimuli combination is presented, therefore does not account for the iterative nature of the experiment.

Normally a simple interaction term in a regression model is usually in a multiplicative nature. Through computational modelling, we demonstrate that the way facial emotion information is integrated is via a parabolic relationship acting on specifically the inequality term.

The way the model captures the iterative nature of the experiment can be explained with an example:

The example below is just given for illustrative purposes and we have explained this process in more descriptive terms in the results section where we make the case for the best fitting model.

Let's imagine that:

acceptance probability for face 5 & offer 350 is .25 and it is rejected by the participant.

The next trial follows with the same offer but the proposer is little bit more unpleasant, let's say face 3 offer 350.

In our model, because the inequality term is modulated by the emotions in a parabolic shape, deviations away from face 5 (or the perceived neutral expression) would lower the negative contribution of the inequality term, and therefore the acceptance probability increases, let's say from .25 to .35.

Let's imagine the participant still rejects this offer and then the proposer's facial emotion gets even more negative while insisting on the same offer amount i.e. 350p. In that case, the acceptance probability according to the model would increase further, let's say from .35 to .47 and this time around (after the proposer insists on the same offer amount with displaying more and more negative faces) the participant accepts the offer. This is why we think the model captures human behaviour under affective load as well as compromise behaviours in the cases where the offer is unfair but the proposer is happier.

If the reviewer thinks that the descriptive interpretation of how the model works does not convey how it accounts for iterative decision-making behaviour clearly enough in the manuscript, we are happy to make further changes to the section so that the readers can fully understand the workings of the model.

5. Pupils

The results seem interesting, just completely unrelated to any other part of the results.

We would like to thank the reviewer for this criticism. In this paper, we endeavoured to use the most common approach in cognitive computational neuroscience in modelling physiological data. We started off with identifying the best fitting model that can account for participant choice behaviour and we used outputs of this model as regressors to analyse the physiological data so that we can marry both of these data modalities and show evidence for physiological processes underlying participant choice behaviour. In doing so we also attempted to regress out as many task components as possible so that we can have a purer signal out of the key model regressors. These are shown in Figure 6 and the equation typed inside the frame completely overlaps with equation 9, which describes the best fitting model. We hope that the reviewer will agree with us that in such a complex paper, we made all the effort to link different components as much as possible. If the reviewer has further recommendations we would be happy to address these in the next iteration.

6. Structure

This paper was not easy to follow. It feels like it was originally written in a 'methods-first' format and hastily transformed to 'methods-last' format. The reader is constantly surprised by unexplained bits of information (the algorithmic vs human proposer is mentioned in one time), new questionnaires, tasks and ratings are presented with no clear indication to their order or any related hypothesis. The demographics are part of the results, presented first, and not a part of the participants section in the methods.

We would like to thank the reviewer for this set of comments. Frankly speaking, this paper had been written in methods last format from the beginning and had been reviewed in few general

audience journals which use that format and was revised few times based on previous feedback. However, we do agree that, sometimes the methods last format usually makes things little bit more difficult to follow, especially the technical aspects of the papers as it prioritises a progression from introduction to results. In computational neuroscience, these technical aspects are very important and this is a common criticism to majority of the papers in this domain (at least based on my reading of some of the decision letters openly published by some general audience journals, a criticism of papers being difficult to follow is fairly frequent). Nevertheless, science in journals with broad readership should be as accessible as possible and we made a sincere effort to improve on these issues pointed out by the reviewer as much as possible (while trying not to change structural aspects of the paper which were praised by other reviewers) and we would be happy to make further tweaks to any outstanding issues, especially if the Reviewer can refer to specific line numbers.

Some parts of this manuscript seem disconnected from one another. The generic introduction about pupils and NE, and indeed the entire pupil diameter analysis, do not integrate with the rest of the paper.

In the revised submission, we aimed to streamline the introduction little bit more to make sure that different components are better integrated. A limited number of previous studies indicated a role for central arousal systems during social decision-making in the UG, and in this study we wanted to explore the workings of the system with further computational model-based analysis approaches. If the reviewer has more specific recommendations as to how to improve the integration of the introduction, we would be happy to make further revisions.

The long discussion regarding FERT (with no mention of what this acronym represents), presentation of stimuli, response time, anxiety and depression seems more suitable in other parts of the manuscript (methods? Supplementary materials?).

We agree with the reviewer that in our initial submission of the paper, some of these components which were not necessarily the key findings of our study and accounted for more of the discussion than needed. These sections are now moved to Supplementary materials under Supplementary Discussion.

The introduction starts with bargaining example of head of state, but it seems weakly connected to the problem at hand (no emotions or facial expressions are mentioned in the example, and there is a lot of literature about intergroup relations and bargaining). Also, stating that there is no information about the computational and physiological mechanisms of iterative social interactions is too broad and wrong – the next paragraph considers such examples, and there are lots of computational models of iterative games (game theory perspective, decision making perspective POMDP), as well as behavioural data (iterative prisoner dilemma, iterative trust games, iterative stag hunt) and neurological data (Autism and UG by Montague for example).

In the introduction of the paper we wanted to start with a generic and more broad example and narrow it down in the subsequent sections where we introduced how facial emotions might also

influence social decision-making. We endeavoured to include many other references that the reviewer and other reviewers recommended, to give a broader account of the previous work being done in this field, which we might have overlooked in our original submission. If the reviewer has any specific papers that s/he thinks should be integrated to the introduction or a better example for the opening of the paper, we are more than happy to make further revisions to the section.

We wanted to focus on nonclinical literature, this is the only reason why Autism paper was not included.

Overall, I think that a lot of editing is needed in order to piece together all the scattered information, to edit out irrelevant information, and to add relevant background works.

We think that thanks to the reviewer's comments the revised submission should be much better in terms of quality and flow over our initial submission and we endeavoured to make as much changes as possible following reviewer's feedback.

Reviewer #2 (Remarks to the Author):

This is my first review of the manuscript COMMSBIO-21-0206 entitled “Dynamic modulation of inequality aversion in human interpersonal negotiations”.

Summary

In this paper, the authors use a repeated version of the Ultimatum game to investigate human negotiation in gradually evolving environments. In this study, the proposer’s facial emotions are expressed and communicated to the receiver before the offer is made and after the respondent decided whether to accept or reject it. The authors propose and test different models to describe possible drivers of the observed human choice behavior. They found that both the proposer’s facial emotions and the amount offered affect the choice behavior of the respondent. In a subsequent model-based analysis the authors found that the decisions of the respondents are guided by aversion to inequality. They found that the best-fitting model is the one that assumes non-linearity in the way the proposer’s facial emotions are perceived and act on participants’ perception of inequality. Finally, the authors recorded pupil dilation and found that pupil size encodes the magnitude of self – other inequality prior to decision onset.

We would like to thank the reviewer for this synopsis of our manuscript, which correctly captures the key aspects of our paper.

General evaluation

The manuscript is written clearly and the research questions addressed in this study are particularly relevant. In particular, the idea to test how trial-by-trial computation of decision values underlying human responder behavior takes place when information about the affective state of the proposer is available before and after the respondent decision is an interesting and challenging research question. Data analyses are conducted in a logical and rigorous manner, and there are many interesting results reported in the paper.

We would like to thank the reviewer for positive feedback on the general organisation of the paper and our analysis approach. We would also like to thank the reviewer for agreeing with our own assessment of the manuscript which we think communicates many interesting findings about human social decision-making. We endeavoured to make further changes to improve on quality and flow of so many different components to make sure that the overall message will remain accessible to a broad readership.

The only main concern and limitation of this research are in the experimental procedure implemented by the authors. The description of the experimental procedure is not particularly clear and leaves several doubts on the real possibility of considering the observed choices of the

respondents as the result of natural social interaction. In particular, I am wondering if the experimental procedure proposed by the authors may have induced the respondents to make choices that are not in line with their real strategic preferences.

We would like to thank the reviewer for highlighting this possibility. In the revised submission we included data from an additional control experiment [In 807] In with overlapping results which we hope will address the reviewer's concerns.

I will try to be more precise in what follows:

- At the beginning of the experiment, participants were told that their proposers would be selecting one offer out of a window of different offers. It is unclear what the participant knows about this window. Does the participant receive information about how the window is selected and the size of the window? From what I understand, by reading the procedure, the participants did not receive any information about this window. In my opinion, this procedure determines different sources of uncertainty for the participants: Is the offer low or high because of the (randomly???) selected window? What is the proposer trying to tell me with this emotional expression? Maybe that he could not offer a higher amount? The authors introduce a source of uncertainty in the respondent decision. If the respondents are unaware of the possible offers available to the proposer, then the participants are not playing a repeated Ultimatum game but a more complex game with private information.

We completely agree with the reviewer's assessment that the instructions for the original experiment might have biased participants to make decisions in a certain way. In our control experiment, we only told participants to make decisions freely and emphasised that in this game there is no right or wrong answer without giving any indication of the opponents potential strategies.

- The authors state that this measure (an offer selected out of a window of different possible offers) was taken to be sure that the decision-making process was confined to responding to combinations of faces/offers in a gradually evolving task environment and the influence of higher-order cognitive processes (theory of mind, learning about the proposer's strategy) is limited or become redundant. Personally, I cannot see why increasing the complexity of the strategic interaction can limit theory of mind processes or reduce the respondent's propensity to infer the rationale behind the decisions of the proposer.

We would like to thank the reviewer for raising this issue. First of all, we removed our previous suggestion that ToM processes would be redundant, this felt too strong as an assertion. In our original experiment, we were reasonably worried about the necessity of recursive modelling and that's why we wanted to give more detailed instructions about the task, in order to minimise these effects. Our, proposed model from the original experiment was able to account for iterative decisions reasonably well (i.e. model predictive accuracy) through an integration of affective information that dynamically modulates inequality aversion. In the control experiment, we did not give any detailed instructions about the task and we told participants to make decisions freely and that there would not be any right or wrong decisions in this game. Considering that our proposed

model was still able to explain this 2nd dataset better than relative competitors (with parabolic family of models still leading the way), we acknowledged that perhaps our initial approach had been overcautious and rather unnecessary. Instead of removing the original section altogether, we added this acknowledgement from the 2nd study so that the reader can also have a window into the evolution of our thinking for this task. If the reviewer wants to have further changes to this section, we would be more than happy to include them.

- The authors told participants that “they may be inclined to accept all offers so that those randomly selected 20 trials would always have a monetary outcome for the participant, but in that case, we told them, that their proposer may detect this tendency and try to make offers as low as possible”. Then they told participants “to use the accept/reject responses strategically to negotiate better terms for themselves.” In my opinion, this is a clear demand effect. This suggestion clearly may have prompted the participants to choose differently from their initial strategic preferences. The observed behavior might be significantly different to the one we would have observed if participants had been free to use their own preferred strategy. This rise some doubt about the generalizability of one of the main result of the paper: the best fitting model, that predict 86% of participants decisions in a repeated Prisoner’s Dilemma, is the one that assumes non-linearity in the way the proposer’s facial emotions are perceived and act on participants’ perception of inequality. We do not know if this model would still be the best fitting model of the natural behavior of the respondents. Similarly, this suggestion might have affected the results of the logistic regression model: “stimuli shown in proceeding trials as well as participants’ previous decisions influenced their choice behavior on the current trial”. On one side, the authors are trying to reduce the respondent’s propensity to infer the rationale behind the decisions of the proposer, on the other, they told participants to be strategic and use accept/reject responses to negotiate better terms for them. If one of the main objectives of the paper is to increase the “ecological validity in terms of capturing real-word human social interactive decision making processes” this cannot be achieved “driving” the behavior of the participants.

This comment also relates to the preceding comment about the complexity of the task which might have been introduced by more specific task instructions given to the initial cohort. But in our control experiment, we did not tell any specific instructions to the participants and told them to make decisions freely, however they liked. In the secondary cohort, we observed fairly overlapping results to the original experiment (also ruling out the effects of performing the rating tasks before the main experiment as in the control experiment these rating tasks were completed after) which seems to suggest that the model that we proposed behaves fairly stable in this kind of iterative social interactions where participants need to integrate the opponent’s affective information with their decision value computations.

We still think we are a few steps away from capturing human social interactions in the most ecologically valid way, that is two individuals interacting freely (a proposer versus a responder) without detailed task instructions. However, to be able to generate a priori hypotheses about how

a free interaction should be modelled we wanted to isolate these 2 steps in 2 papers (e.g. Pulcu and Haruno, 2019 investigating the proposer behaviour and demonstrating the suitability of risky decision-making models, whereas the current paper showing how inequality aversion can be dynamically modulated to account for responder behaviour). We hope that we will get to that stage where we will be able to test these predictions in our future work.

Other comments

- Lines 102 - 104: provide a description, somewhere in the paper, about how these sliding windows with transitions probabilities work.

Thank you very much for highlighting this omission. We referenced to the supplementary materials where more information is given about the strategy of the opponent and we further added some descriptive information to the Procedures section to make sure that readers can have an idea about how the experimental paradigms worked.

- Lines 287 – 292: The opponent-type regressor which determines whether the participants played against a human or a computerised opponent did not have any significant main effect on human behaviour or parameter estimates. It is possible that recent advances in AI mimicking, and even excelling, human behavior in competitive games might have an indirect effect on these results. An alternative hypothesis is that the choice behavior of the participants is in both cases (human vs computerized opponent) simply following the “suggestions” provided by the experimenters about the decision strategy that they should implement. This would reduce significantly possible differences in the two conditions.

Thank you very much for highlighting this possibility. We think that our control experiment which was free of all of the detailed instructions which were given to the original cohort, addresses this issue. In the control experiment, participants knowingly played against the computerised strategy (i.e. no deception or debriefing was used, this approach was necessary in order to do laboratory experiments while mitigating coronavirus transmission risk) and we report qualitatively very similar results to the original experiment in Supplementary Figure 8.

- Lines 457 – 464: the authors state that there is no work prior to the present study that has on pharmacological manipulation of the central NE system and how this would affect perceived inequality in humans. This part is confusing; in this study there is no pharmacological manipulation.

We agree with the reviewer and we removed this sentence altogether.

- In general, the discussion is too long and some parts are not strictly linked to this research work. In particular, the second part of the discussion can be significantly shortened.

We would like to thank the reviewer for his/her critical appraisal and we moved one complete section of the discussion to Supplementary discussion in an attempt to shorten the main body of the manuscript. We would like to thank the reviewer again for all the recommendations which we think substantially improve the quality of the manuscript over its initial submission.

Reviewer #3 (Remarks to the Author):

The manuscript by Murphy et al. describes how humans behave in a version of the ultimatum game which allows studying the interplay between the proposer's facial emotions and the offer amounts. To review the paper, I focused on the main criteria displayed in the journal's website (<https://www.nature.com/commsbio/about/aims>). In my opinion, while this paper shows additional evidence that emotions affect economic choices, I do not believe that the results reported here represent "an advance in understanding which may influence thinking in the field" (including the sub-field of human economic decision making).

We would like to thank the reviewer for this initial assessment. We endeavoured to make necessary changes to address the reviewer's concerns about the caveats of our work, as well as highlighting its novelty.

NOVELTY OF THE RESULTS

The novel contribution of the paper compared to the existing literature remains unclear. In the Introduction, the authors make several strong claims about novelty which are either overly specific (point 1) or too general to be true (point 2):

In our evaluation, our current paper makes a significant novel contribution to the field by showing that during social interactive decision-making human participants integrate their opponent's affective information with their value computations. Although inequality aversion models had been proposed for UG, no one has proposed a model about how these models can be further improved to account for how affective information is integrated. In that respect, we specifically evaluated the suitability of exponential models which were proposed for human reinforcement learning and we identified that parabolic models actually account for human social decision-making much better. We also show that the parabolic modulation specifically applies for the inequality term in the best fitting model rather than self reward amount or both. We think that, the parabolic assumption should also be tested for human reinforcement learning data in the future (e.g. an early implementation from the corresponding author: Pulcu, 2019, a nonlinear relationship between prediction errors and learning rates, <https://www.biorxiv.org/content/biorxiv/early/2019/09/29/751222.full.pdf>). We also conducted a model-based analysis of the pupillometry data to demonstrate that pupil dilation selectively encodes the magnitude of the inequality between the proposer and the responder. To the best of our knowledge, these results (including modelling facial emotion ratings which were secondary to

our research question) were not previously reported in the literature, highlighting the novelty of the current work.

1) Lines 83-84: “how trial-by-trial computation of decision values underlying human responder behaviour takes place in an iterative UG with an ecologically valid affective component has not been shown before”. There are several papers that have used computational approaches to analyse trial-by-trial variability in iterative UG and reported more than just “summary statistics of average acceptance probabilities” (line 82). For example, Gabay et al. (2018) used a statistical model to study the psychopharmacological effects of MDMA and psilocybin on responder behaviour in the UG at the trial-by-trial level (<https://www.nature.com/articles/s41598-018-26656-2>). Si et al. (2020) implemented a decoder model that successfully predicts responder behaviour using trial-by-trial variability in the EEG signal (<https://www.sciencedirect.com/science/article/pii/S1053811919309243>). Xiang et al. (2013) even used model-based fMRI and implemented an ideal-observer computational model to show how signatures of trial-by-trial computations of decision value are encoded in the striatum, insula, and OFC (<https://www.jneurosci.org/content/jneuro/33/3/1099.full.pdf>).

While none of these studies have an “ecologically valid affective component”, this can also be said to be the case in the present study (where do we set the threshold for “ecological validity”?). And even if one believes that this study has an ecologically valid affective component, then the main contribution of the paper would be simply extending a widely used approach to a more ecological setting, something that is unlikely to represent a significant advancement in the understanding of economic decision-making.

2) Lines 140-141: “the role of pupil-linked central arousal systems during social interactive decision-making remains unknown”. The claim here is simply too general to be true (social decision making includes, for example, the analysis of moral choices which have been studied with pupillometry). In fact, there is even one study that has linked pupil size with participants’ behaviour in the UG (Rago et al. 2015, <https://www.tandfonline.com/doi/abs/10.1080/17470919.2015.1057295>).

Overall, this left me unconvinced that the paper makes a contribution to the literature that is both meaningful and novel. Moreover, there are other papers that have shown evidence for the main conclusion of the paper (“inequality aversion is a malleable cognitive process”). For example, Vavra et al. (2018) showed how simply changing people’s expectations about what constitutes a standard offer modifies people’s behaviour in the UG. One could say that this paper already showed that aversion to inequality is a malleable process (<https://www.frontiersin.org/articles/10.3389/fpsyg.2018.00992/full#B24>).

We would like to apologise for the omission of these literature from the original submission. In the revised submission, all of these papers are cited and mentioned in different sections of the

manuscript, some of them were more suitable for the introduction whereas others more suitable for the discussion section of the paper. Of these recommendations, 2 papers are worthy of mention more in detail, both of which made important contributions to social decision-making literature but used a one-shot design instead of an iterative design as in our current work.

Xiang et al., 2013 provided a model of social decision-making in the UG, but this model focused on one-shot decision-making coming from different proposers drawn from a population. Prior to social decision-making participants learn about the features of these populations such that they would build an expectation about what kind of offers they would get, allowing their modelling to rely on a reinforcement learning framework and prediction errors.

Vavra et al., 2018 used a similar design in which participants interacted with proposers over one-shot UG offers where there were 4 different populations/distributions of offers. In this scenario, their work demonstrated that about half of the population systematically adjust their behaviour to the statistics of the environment, whereas our work shows how inequality aversion is malleable in relation to affective information. To further highlight the novelty of our approach, one can give an analogy from reinforcement learning.

Vavra et al's work would be similar to using a Rescorla-Wagner model to assess participants' learning rates in different environments and demonstrate that learning rates change when the environment changes (e.g. as in Behrens et al., 2007, Pulcu & Browning 2017). However, this assessment approach does not make any mechanistic proposals about how learning rates change dynamically. In RL literature Pearce-Hall family of models make an assumption about how learning rates should change as a function of prediction errors, consequently making a strong and novel contribution to that field. Our proposed model in this paper is similar to the latter, that we make a proposal about how affective integration models can account for social decision-making in iterative scenarios by flexibly tuning inequality aversion. We hope that this analogy further highlights the novel contribution of our paper, which we think was not shown before.

We agree with the Reviewer that the highest ecological validity in these tasks could be achieved by a two-person free interaction scenario. Previously, we focused on the proposer behaviour (Pulcu and Haruno, 2019) and in the current work we wanted to isolate and model the responder behaviour. In our future work, we are planning to start conducting two-person social interaction studies. Our a priori hypotheses based on these ongoing work would be that human interpersonal negotiations can be explained by an interplay between risky decision-making and inequality aversion models, but this is a prediction that needs to be tested explicitly in future work.

EVIDENCE FOR THE CONCLUSIONS

1) The downside of breaking trial-wise independence. One of the main features of the paper is that the task uses an "ecologically valid" experimental design. However, participants know they are interacting with an avatar and so their behaviour might be completely different to the one observed

in real pairwise interactions between individuals. In fact, one caveat of breaking inter-trial independence (to gain in ecological validity) is that the authors are also breaking the statistical independence between stimuli presented at different trials, which are used as different predictor variables in a multivariate regression. The paper does not provide any evidence that this approach is justified and that there is sufficient unshared variance between consecutive stimuli (what is the variance inflation factor of the multivariate regressions?)

We would like to thank the reviewer for raising this possibility. In our original submission we investigated the degree to which regressors for the pupillometry analysis were not correlated, and previously reported these results in Supplementary Figure 6A. For other multiple linear regression models, we had anticipated a correlation between depression and anxiety scores in this cohort of healthy volunteers and in order to overcome any problems arising from this, we have computed a composite index of depression and anxiety. We think this may be warranted because, in this manuscript our participant cohorts were always nonclinical volunteers and the relationship between depression and anxiety scores were not our primary research question, we attempted to document human decision-making work as detailed as possible to inform future research. In the revised submission, we computed the variance inflation factor for the multiple linear regression models that we used. The VIF were in the desirable range around 1 and this is reported in Figure 2 legend and applies to all other multiple linear regression models reported in the paper as we used the same model/regressors to explore the relationship between model parameters and questionnaire measures and demographics. We also report a similar VIF factor values for the logistic regression model recommended by Reviewer 1, revealing similar values.

2) No evidence for an effect vs. evidence for no effect. Figure 2B is supposed to convey the message that the model parameters fitted in Eq. 1 are not influenced by demographic factors and symptoms of depression/anxiety. The evidence presented for this conclusion is clearly insufficient. Showing that regression coefficients are within an interval including zero (i.e., showing that beta values are n.s.) might simply mean that the study is underpowered to detect those effects (something that could be the case given the large C.I.). In other words, the study shows no evidence for an effect, and the authors report this as evidence for a null effect.

We completely agree with the reviewer, that our current work is not a definitive study in terms of its sample size. The relationship between constructs such as depression and anxiety scores are better suited for future clinical or sub-clinical research in which one can specifically recruit people with high and low symptoms in order to establish whether there is any systematic relationship between these external symptom scores and model parameters derived from participant choice behaviour. These are now acknowledged in the paper [In 339].

TECHNICAL SOUNDNESS OF THE DATA

Line 111: “Our a priori hypothesis was...” With the data presented in this paper, the reader cannot be confident enough that the presented data tested “a priori” hypotheses since there is no pre-registration of the hypotheses, methods, or analyses. In fact, some passages of the paper suggest that results went in an unexpected direction. For example, the authors report a rather counter-intuitive non-linear relationship between actual and perceived emotion (line 170: “This bimodality indicated the possibility of a nonlinearity in human facial emotion recognition (...). To capture this nonlinearity,...”) This suggests that a main feature of the task (perceived emotion) follows a complex rule that was added ad-hoc to explain a pattern that was unexpected by the authors. While there is nothing inherently wrong with this situation (empirical science often goes in unexpected ways), the authors should be crystal clear about which of the hypotheses and analyses were exploratory and which of them were confirmatory. Moreover, if one or more of the presented models were selected in a post hoc data-driven manner, it would be reassuring for the reader that the results replicate in a second dataset (ideally pre-registered).

We agree with the reviewer that more information can be communicated with the reader to highlight the a priori hypotheses versus analysis which are conducted in an exploratory fashion. First of all, the task which was used in this paper was described in one of our earlier publication in Lancet Psychiatry (Pulcu and Elliott, 2015) and we have been working towards implementing that vision in the past few years. The rating task specifically focusing on facial emotion recognition was intentionally based on a 1-9 Likert Scale, which would make it easy to exploit the properties of the probability weighting function to capture such perceptual biases as our ongoing and previously published works (e.g. Pulcu and Haruno, 2019; Pulcu, 2019) demonstrated the versatility of these functions apply to other domains also including reinforcement learning. We were not surprised to find such non-linear relationship in our data, but obviously one cannot have an a priori expectation about the exact trajectory of these functions without having any previous literature to generate hypotheses. In the revised submission, we explicitly stated which analyses were exploratory and which were strictly hypothesis testing.

It is worthwhile to highlight that the models were selected based on Bayesian Information Criteria in a data driven measure, at least for the cohort reported in the initial submission. In line with comments from multiple reviewers, we collected an independent cohort of participants and try to address some of the limitations of the original experiment which might have originated from task instructions or the time point in which the rating tasks were completed. Although, we did not preregister this secondary control study (which involved a lot of paperwork due to COVID19 risk mitigation), we did have an a priori expectation to fully replicate our modelling results and the results from the control experiment suggested that the model that we identified in a data driven manner in the 1st experiment can be generalised to account for human behaviour in an independent cohort.

We hope that the revisions that we made in the revised submission accurately addresses the concerns that the reviewer had about the original submission. We would like to thank the

reviewer again for all the recommendations which we think improve the quality of the paper substantially over its initial submission.

Reviewers' comments:

Reviewer #1 (Remarks to the Author):

I thank the authors for their responses. I really like the approach in this work, and the notion of exploring how emotional signals shape economic decision making. I am satisfied with the replication of the experiment without too detailed instructions, and without ratings before the experiment. The authors also revised the pupil-size section, although it still seems like a more exploratory aspect of the paper and should be stressed that it is so.

However, following the authors clarifications regarding modelling, it now becomes clear that some very serious confounds in the experimental design, which make some of the conclusions hard to justify.

The most important is that the experiment is sequential, i.e. each step directly influences the next step, both in terms of facial expression and in terms of the proposed split, but this aspect is ignored by the authors, and their models assume independency between steps.

For example, the winning model assumes that likelihood of acceptance of an offer increases with deviance of facial expression from neutral. They provide an example, both in the rebuttal and the paper, where this can explain a trajectory where the same offer is being made with facial expression that keeps getting more and more negative. At some point this negative face is so far from neutral, that it overcomes the inequality of the offer and the participant accepts. However, the model is completely indifferent to this trajectory. It should make the same prediction if a rejected offer was repeated with more and more positive face (as it is parabolic), until one sees a very happy face with very bad offer and decides to take it. This trajectory is very strange, and could never occur in the task – as positive faces mostly come after acceptances. This illustrates one problem in the task – facial expressions are tightly connected to previous acceptance/rejections and to previous proposals, and therefore it is almost impossible to interpret any model predictions beyond this relationship. Another strange prediction would be lowering the likelihood to accept offers once a facial expression accompanying them becomes neutral.

Another problem is the lack of control conditions. What if facial expressions were absence? In this case it is possible that simply repeating the same bad offer over and over again will result in acceptance (i.e. the participant realizes that he should accept a low offer or risk not getting anything). This trajectory is exactly the same as before, but without any facial expressions. If it is possible to replace facial expression with number of previous rejections (which is exactly how the faces are generated in the task), this suggests that there is no effect of facial expressions.

Another control condition could be a random facial expression – or coupling them with different types of rules (such as more positive after rejection) to disentangle the effect of facial expression on acceptance.

Overall the lack of any control condition makes the conclusions of this study very shaky.

The authors include a logistic regression which models the current decision with current face, and previous faces and offers. This seems strange – why not include current offer as well?

In any case, the logistic regression results in Figure 4 show a clear dependency on previous offer, above and beyond the current facial expression – which indicates that it is not facial expressions that play a role, but the persistence of the offer.

The description of results obtained from the new cohort of participants is sparse – why not show the same results (logistic regression) as the first experiment?

Reviewer #2 (Remarks to the Author):

This is my second review of the manuscript COMMSBIO-21-0206A entitled: Dynamic modulation of inequality aversion in human interpersonal negotiations.

General Evaluation

In general, I like the improved version of the manuscript and I appreciate the effort the authors put into it. In particular, the (new) control experiment solves the most critical aspects of the previous version of the manuscript.

I would like to thank the authors for answering all my questions and for producing a revised manuscript in line with my questions and suggestions.

Reviewer #3 (Remarks to the Author):

I would like to thank the authors for addressing all my previous concerns. I have no further comments.

Reviewers' comments: **[responses to the Reviewer are typed in bold]**

Reviewer #1 (Remarks to the Author):

I thank the authors for their responses. I really like the approach in this work, and the notion of exploring how emotional signals shape economic decision making. I am satisfied with the replication of the experiment without too detailed instructions, and without ratings before the experiment. The authors also revised the pupil-size section, although it still seems like a more exploratory aspect of the paper and should be stressed that it is so.

We thank the reviewer for his/her positive feedback on the amount of effort that we put into revising the manuscript. In the current iteration, we put equal if not more effort in revising and improving the manuscript in the light of the Reviewer's guidance. We highlighted that we explored how pupil size correlates with the regressors from the best-fitting computational model (for example, lines 156, 381, 465).

However, following the authors clarifications regarding modelling, it now becomes clear that some very serious confounds in the experimental design, which make some of the conclusions hard to justify.

The most important is that the experiment is sequential, i.e. each step directly influences the next step, both in terms of facial expression and in terms of the proposed split, but this aspect is ignored by the authors, and their models assume independency between steps.

If our understanding is correct, the Reviewer suggests that our models assume independency between the steps, probably because modelling the choice behaviour focuses on the task inputs from the current trial. However, because the current trial inputs are generated based on previous history, it would absorb most of the information and it would not be necessary to model the complete task history. Similar assumptions (e.g., Markovian) are commonly made in social interactive decision-making tasks like Prisoners Dilemma, or in simple (e.g., Rescorla-Wagner) and Bayesian models of reinforcement learning (i.e., modelling the full history of the interactions with the environment would only have negligible improvement over models which focus on the current or one preceding trial, consequently this is not performed). This is now mentioned in the Methods section, line 720.

For example, the winning model assumes that likelihood of acceptance of an offer increases with deviance of facial expression from neutral. They provide an example, both in the rebuttal and the paper, where this can explain a trajectory where the same offer is being made with facial expression that keeps getting more and more negative. At some point this negative face is so far from neutral, that it overcomes the inequality of the offer and the participant accepts. However, the model is completely indifferent to this trajectory. It should make the same prediction if a rejected offer was repeated with more and more positive face (as it is parabolic), until one sees a very happy face with very bad offer and decides to take it. This trajectory is very strange, and could never occur in the task – as positive faces mostly come after acceptances.

We agree with the Reviewer that, it is less likely that the proposer would display increasingly positive faces if the previous offers were continuously rejected and if the proposer insists with the same offer (i.e., if the offer amount stays as a constant). For example, on average only 7.6% of the trials involved a more positive face upon rejection of an offer, but here it is important to understand that the rejection of the offer is also depends on participants inequality aversion. The parabolic influence of the faces tweaking the inequality term would also depend on the strength of inequality aversion for a given individual. For example, if the inequality aversion is very strong, positive or negative faces increasing the acceptance probability from let's say 20% to 25% would not necessarily mean a behavioural switch, i.e., from reject to accept (0 to 1). This is now highlighted from line 285. Finally, although we name this influence as parabolic for the ease of reader's understanding, the exact trajectory of the affective influence would depend on the steepness and nonlinearity of the curve accounting for perceived emotions at single subject level as shown in Figure 2B. Example of single subject trajectories are shown below:

Nevertheless, some elements of the parabolic trajectory that we proposed can be seen even in the model-free summary heat map, where the acceptance probability is higher at the polar extremes of facial emotions and there are dips in between (e.g., Figure 3A, vertical trajectory from top to bottom when offers are between ~350-600 on the x-axis). However, we do acknowledge that we did not have too much power to truly dissociate the effects of facial emotions alone because, irrespective of participant decision, the longest sequence of trials experienced by participants in which an offer stayed the same was on average 3.5 trials (min-max: 2-11). In the current iteration, we acknowledged this as a

limitation and as a recommendation for future research to investigate more thoroughly (from line 495).

In our opinion, certain conditions not happening frequently in the task does not take away the flexibility of the model to adapt to different situations (i.e. the whole stimuli space and considering each single subject has their own non-linear trajectory of facial emotion recognition and inequality aversion) as we think it is this feature that allowed the model to explain participant choice behaviour much better than other competing models that we considered, along with its good predictive accuracy, ability to recapitulate the raw data and having desirable generate-recover features. We think the most important aspect of the model is its flexibility with which it can adapt to different situations that can occur within the whole stimuli space.

This illustrates one problem in the task – facial expressions are tightly connected to previous acceptance/rejections and to previous proposals, and therefore it is almost impossible to interpret any model predictions beyond this relationship.

We would like to highlight that facial expressions were probabilistically connected to the previous outcomes as, at least in our opinion, would better reflect real life social exchanges such that a proposer would be more likely to display a happier face relative to the previous point if their offer is accepted. We wanted to highlight this feature of the task as tightly connected may sound like a deterministic association.

Another strange prediction would be lowering the likelihood to accept offers once a facial expression accompanying them becomes neutral.

Here, we would like to clarify that the prediction of the model is that when the faces are neutral people would stick with their inequality aversion and self-reward terms which would guide their choice behaviour under the influence of the offer amount. As highlighted earlier, there are parts of the stimuli space where the acceptance rates were higher at the polar extremes of facial emotions relative to neutral.

Irrespective of the explanations that we provided in this section, if the Reviewer has a specific model in mind which should apply better to the data that we presented, we would be more than happy to test that model against our proposal and revise the paper accordingly.

Another problem is the lack of control conditions. What if facial expressions were absence? In this case it is possible that simply repeating the same bad offer over and over again will result in acceptance (i.e. the participant realizes that he should accept a low offer or risk not getting anything). This trajectory is exactly the same as before, but without any facial expressions.

Here, the Reviewer raises an interesting question. Although it was not possible to conduct

an additional experiment (due to unavailability of research funds and other necessary resources) or ask for data sharing from another research group so that we can test this hypothesis on an existing dataset, it was still possible to test whether our participants approached the experiment with a “risk of missing out” framework described by the Reviewer.

In that case, the participants expected outcome at the end of the experiment would be a function of their average acceptance rate multiplied by the average offer they received during the experiment (i.e., as in multiplicative expected value). Considering that 20 trials selected at the end of the game for additional reimbursement were selected purely randomly, participants could never know which trials would be selected during the game, therefore they would need to rely on this expected outcome calculation. This would also mean that the influence of faces as we proposed must sway participants away from this calculation such that there should be a negative relationship between the fitting of the parabolic influence model and the expected outcome computed according to risk of missing out. Indeed, there is a negative relationship between these metrics ($r=-.21$, $p=.17$) in the main experiment ($N=44$). This relationship would be significant if we combine both of the cohorts together ($N=69$) considering the results from the control experiment were reasonably overlapping ($r=-.29$, $p=.016$). We think that former lack of significance might as well be a power issue as our experiment was not designed to investigate this relationship specifically. These complementary results are now reported in the paper as exploratory findings, from line 325.

If it is possible to replace facial expression with number of previous rejections (which is exactly how the faces are generated in the task), this suggests that there is no effect of facial expressions. We are not 100% sure why the Reviewer thinks there is no effect of faces on the choice behaviour. For example, if the participants approached the game with a risk of missing out mindset, to make sure that all of those randomly selected trials had acceptance outcome, such that the proposed amounts on those selected trials would contribute to their additional reimbursement they would be accepting all offers. This would also mean that there should not be any main effect of offer amount either, i.e., the participants would be insensitive to the changes in the offer amount and continue accepting all offers (rejecting all offers would also mean insensitivity to offer amount). However, our findings in Figure 3 (the heatmap and regression coefficients), clearly demonstrate that both of these factors (facial emotions and offer amount) influence human choice behaviour. Particularly the heatmap in Figure 3A relies on raw averages directly from participant decisions and therefore immune to any possibility that model assumptions may be wrong. It is also important to highlight that models without any facial emotion component did not fit the participant choice behaviour well (e.g., models based on offer liking ratings or online

decision values). However, our results do not necessarily mean that all participants will take the influence of faces into account at the same level either. For example, a strong inequality aversion would also yield insensitivity to the emotional influence. We provided further clarifications in the manuscript highlighting this possibility (from line 285).

Another control condition could be a random facial expression – or coupling them with different types of rules (such as more positive after rejection) to disentangle the effect of facial expression on acceptance.

Overall the lack of any control condition makes the conclusions of this study very shaky.

To the best of our knowledge, there is no previous study in the literature which used an iterative ultimatum game paradigm in which the offers were generated probabilistically based on previous responses, but faces were generated randomly (as we referenced the existing literature more carefully in the 1st round of revision and did not come across any studies fitting this description). Sadly, it was not possible for us to conduct another experiment fitting this description either.

Overall, in this paper we proposed a general and flexible model that can account for iterative ultimatum game behaviour with facial emotions involved and to achieve this we also used a number of different confederates so that we can minimise any possibility of overfitting. This would allow labs around the world to replicate or further test our proposal in similar designs without necessarily relying on identical facial emotion information. Although we were not able to conduct an additional control experiment, we highlighted the limitations of our work and that these models remain as our up-to-date hypotheses, as we mentioned in the closing sections of our discussion (line 503). Future studies should endeavor to tackle these questions in tasks with different rules and ideally in real 2-person interactions.

The authors include a logistic regression which models the current decision with current face, and previous faces and offers. This seems strange – why not include current offer as well?

We thank the Reviewer for raising this question. We realised that we made a mistake by not including the current offer in the revised logistic regression analysis. We agree with the Reviewer that the model that we reported in the 1st round of the revision indeed became strange and we sincerely apologise for this mistake.

In our original submission, we wanted to provide results from 3 complementary analyses, i.e., a linear regression model which demonstrates the main effects on average acceptance probability, a logistic regression model which demonstrates the effect of previous trials on the current choice and a model-based analysis demonstrating how task components such as facial emotions and the offer amounts are integrated into decision values. We now mentioned this complementary analysis logic explicitly in the paper. (line 247).

In the context of social interactive decision-making, a logistic regression model focusing solely on the effects of previous trials would be particularly interesting because in games such as iterative Prisoners Dilemma, agents' behaviours are usually defined with respect to their response to one previous trial (as highlighted at the end of that section, line 255). In order to keep the complementary nature of different types of analyses, we decided to exclude the task components from the current trial in the revised logistic regression results (Figure 4).

Nevertheless, in order to respond to the Reviewer's point, we still would like to present how the logistic regression results would look like if we included all the components from the current trial. Not surprisingly, current offer amount would capture a lot of the effects, deeming the effect of previous offers almost non-existent (results from the main experiment, N=44). It is also worthwhile to highlight that, in this model shown below we did not include any interactions between the current trial components and previous trials as there are a lot of combinations one can come up with (i.e., interaction terms were only defined within each trial but not across). We thought exploring these complex interactions (e.g., face (n-1) x offer (n-2) or (face x offer (n-1) x choice (n-2) etc.) may not be informative as it becomes difficult to construct and interpret such regressors/coefficients.

Although we think including the current trial components are not so informative in the logistic regression with respect to our analysis logic explained above, if the Reviewer still thinks this model would be better, we are happy to put the figure above in the main paper. However, we think this would be conflicting with the main model-free analysis (i.e., Figure 3B, conducted on average acceptance probability rather than binary choice in the logistic model) and will miss the opportunity to isolate the effect of only the previous trials as we intended. We think that the results shown above would mean looking into the effects of the current trial components while controlling for the task history, which would not reflect the spirit of the task (i.e., having

probabilistic transitions between the states).

Implementing the logistic model in a way that overlaps with the linear model we reported in Figure 3B would also reveal similar results (i.e., main effects of faces and offers, along with an interaction, shown below). Therefore, we do not think our results currently shown in Figure 3B can be an artefact of leading this analysis with a linear rather than a logistic model. Arguably, the linear model would be much more sensitive to identify and confirm these effects because data consists of average acceptance probabilities which is shown to have a good gradient as in Figure 3A heatmap.

In any case, the logistic regression results in Figure 4 show a clear dependency on previous offer, above and beyond the current facial expression – which indicates that it is not facial expressions that play a role, but the persistence of the offer.

We partially agree with the Reviewer’s interpretation of the [previous] logistic regression results, negative regression coefficients on the effect of previous decisions that we observed in the main experiment (which had a larger sample size) highlights the influence of previous rejections on current acceptance. This relationship between previous rejections and the current choice also emerges in the updated Figure 4. However, in our opinion neither the effect of previous decision nor the effect of the previous offers equates to persistence of the offer as the offers may be improving (e.g., from 50 to 350 etc.) yet the participant could still reject them if they are not compatible with the participant’s perception of inequality. Therefore, we do not agree that these effects can take away from the effect of faces as shown in Figure 3. For example, the revised logistic regression results (Figure 4) suggest that participants would be more likely to accept an offer if the facial emotion in the previous trial is relatively more positive and if the offer on the previous trial has been rejected. Both of these factors are significant going 2 trials back

and decay further down the trials, following a logical trajectory. We included this interpretation in the main body of the article (ln 291). We also flagged up that across 2 studies (N=44 versus n=25 control experiment), the only regressor which makes a consistently similar contribution is the offer amount (n-1). This is mentioned in line 1331. We decided not to concatenate the data from 2 cohort to refit the model in an effort to simplify the paper.

The description of results obtained from the new cohort of participants is sparse – why not show the same results (logistic regression) as the first experiment?

We apologise for this rather unintended omission. In our 1st revision of the paper, we were worried with how the paper was expanding with respect to content and number of supplementary figures etc. and that is why we only included the summary model-free results from the control experiment and the goodness of fit metrics for model fitting just to highlight that task instructions and the order in which the rating tasks were completed did not influence the results to a significant degree. In line with the Reviewer's recommendation, in the current revision, we also included the figure from the logistic regression analysis for the control experiment cohort (identical to the model in Figure 4), which is now shown in Supplementary Figure 10.